# Beyond the Final Answer: Evaluating the Reasoning Trajectories of Tool-Augmented Agents

**Wonjoong Kim** [1]  **Sangwu Park** [1]  **Yeonjun In** [2]  **Sein Kim** [2]  **Dongha Lee** [3]  **Chanyoung Park** [1 2]

## Abstract

Although recent tool-augmented benchmarks involve complex requests, evaluation remains limited to answer matching, neglecting critical trajectory aspects like efficiency, hallucination, and adaptivity. The most straightforward method for evaluation is to compare an agent's trajectory with the ground-truth, but annotating all valid ground-truth trajectories is prohibitively expensive. In this manner, we introduce TRACE, a reference-free framework for the multi-dimensional evaluation of tool-augmented LLMs. By incorporating an *evidence bank* which accumulates knowledge from preceding steps, TRACE assesses an agent's reasoning trajectory effectively. To validate our framework, we develop a new meta-evaluation dataset with diverse and flawed trajectories, each labeled with multi-faceted performance scores. Our results confirm that TRACE accurately evaluates complex trajectories even with small open-source LLMs. Furthermore, we apply our method to evaluate the trajectories that agents produce while solving tool-augmented tasks, presenting previously unreported observations and their corresponding insights.

## 1. Introduction

With the recent advancements in Large Language Models (LLMs) (Zhao et al., 2023; Chang et al., 2024; Achiam et al., 2023), there has been a surge of interest in tool-augmented agents that overcome their intrinsic limitations by leveraging external tools (Yao et al., 2023; Gao et al., 2025; Qu et al., 2025; Zhang et al., 2024a; Gou et al., 2024; Qin et al.,

2024; Yuan et al., 2025; Li et al., 2024a; Wang et al., 2025). As this field rapidly evolves, various benchmark datasets have been introduced to measure agent performance. However, while extensive evaluation has focused on what tasks an LLM agent can accomplish, the assessment of *how* it accomplishes them has been largely overlooked.

Existing benchmarks predominantly rely on an *Answer Match* evaluation, which only verifies if the final output matches the ground-truth answer (Huang et al., 2024; Wang et al., 2024; Ma et al., 2024; Mialon et al., 2023; Zhang et al., 2024b). However, we argue that agents reporting the same accuracy may not necessarily possess the same level of performance. For instance, two agents might achieve the same accuracy, yet one may arrive at the solution efficiently in minimal steps, while the other might exhibit poor reasoning quality, such as engaging in redundant processes or generating hallucinations. Previous research has largely overlooked crucial attributes that an agent should possess; An agent should be able to construct an efficient reasoning trajectory, avoid hallucination, and demonstrate adaptivity by selecting alternative tools when a chosen one is outdated, version-incompatible, or otherwise unusable. Figure 1 shows cases where two agents arrive at the same result given the same task but through different trajectories, highlighting differences in efficiency, the presence of hallucination, and adaptivity of the agent. As this example illustrates, if we were to report only the final answer accuracy, the two agents would be considered of equal quality. However, an analysis of their trajectories reveals that this is not the case. Therefore, to reliably evaluate an agent's performance, it is crucial to inspect the reasoning trajectory, not just the final answer.

While human evaluators offer the most accurate method for evaluating an LLM's trajectory, the increasing length and complexity of LLM responses have led to a growing reliance on LLM-based evaluators (Bai et al., 2023; Zhuge et al., 2025; Fu et al., 2024; Li et al., 2024b; In et al., 2025; Ling et al., 2023). However, a straightforward approach of using an LLM to evaluate an agent's trajectory by comparing it with a ground-truth trajectory poses a new challenge: user requests can often be solved through multiple valid trajectories. For instance, a recent benchmark dataset (Wang

[1]Graduate School of Data Science, KAIST, Daejeon, South Korea [2]Department of Industrial and Systems Engineering, KAIST, Daejeon, South Korea [3]Department of Artificial Intelligence, Yonsei University, Seoul, South Korea. Correspondence to: Chanyoung Park <cy.park@kaist.ac.kr>.

*Proceedings of the $43^{rd}$ International Conference on Machine Learning*, Seoul, South Korea. PMLR 306, 2026. Copyright 2026 by the author(s).

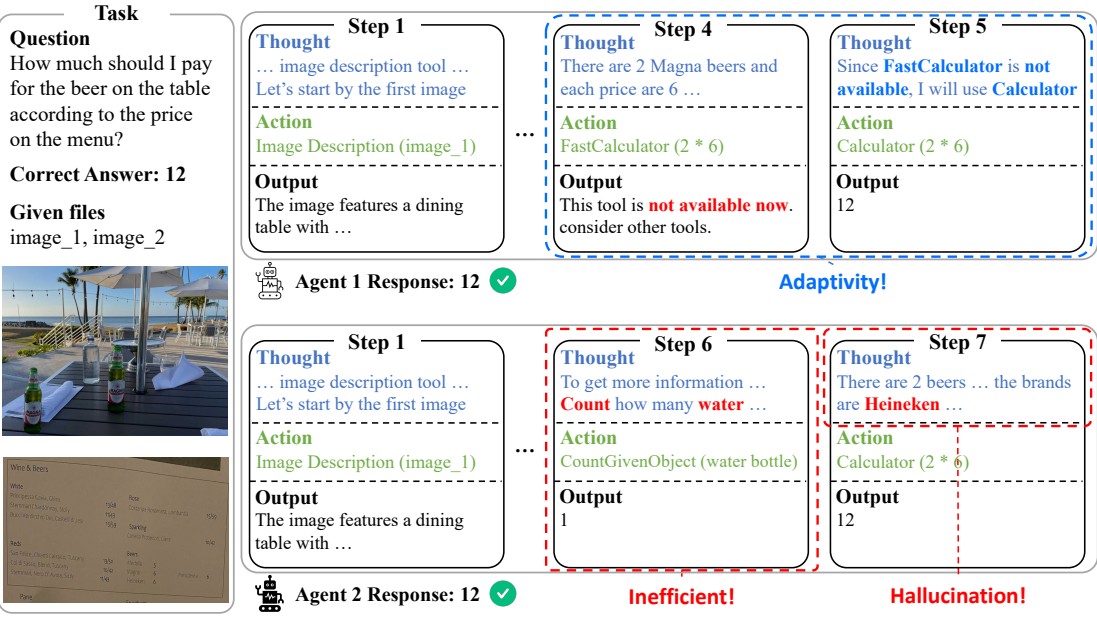

*Figure 1.* An example of agents returning the same answer through different trajectories given the same task.

et al., 2024) provides a single ground-truth trajectory for each user's request, yet we observe that multiple, alternative valid trajectories exist for the same request. Consequently, evaluating an agent's trajectory against a pre-defined, single ground-truth can negatively impact the accuracy of the assessment, but annotating every possible trajectory is prohibitively expensive. Therefore, we need an LLM evaluator that can assess trajectories without ground truth, but a simple LLM evaluator struggles to perform this task effectively, as studies have shown that their performance degrades when evaluating long and complex contexts (Das et al., 2024; Zhang et al., 2025; Krumdick et al., 2025).

To address these issues, we propose **T**rajectory-based **R**easoning **A**ssessment and **C**omprehensive **E**valuation, TRACE, a simple yet effective methodology for in-depth evaluation of a tool-augmented LLM agent's reasoning trajectory. TRACE is a general and effective LLM-based evaluation framework capable of assessing whether an agent has followed a logically sound reasoning process. Incorporating an *evidence bank*, which accumulates knowledge gathered from each reasoning step, TRACE provides a comprehensive understanding of an agent's performance beyond binary success or failure by holistically evaluating multi-dimensional metrics, i.e., efficiency, hallucination, and adaptivity.

To validate our proposed methodology, we construct a dedicated dataset for meta-evaluation. We augment existing state-of-the-art tool-augmented agent benchmark datasets, GTA (Wang et al., 2024) and m&m's (Ma et al., 2024), with diverse reasoning trajectories. These augmentations include cases of unnecessary tool use, hallucinations, and attempts

to select alternative tools following an initial failure. Each augmented trajectory is then labeled with corresponding scores for inefficiency, hallucination, and adaptivity. Using this dataset, we demonstrate that our evaluation methodology accurately assesses each trajectory against these nuanced metrics even with small open-source LLMs.

Lastly, we evaluate the trajectories returned by various real-world LLM agents on tool-augmented tasks using TRACE. We find that even agents previously reported to have similar accuracies in fact exhibit significant performance differences when their reasoning trajectories are examined. From this experiment, we present a range of observations and insights, briefly identify the causes of poor performance in tool-augmented agents, and propose potential future research directions for enhancing their capabilities.

In summary, our main contributions are as follows:

- **An effective and efficient trajectory-focused evaluation framework** that assesses the logical soundness of an agent's reasoning process without depending on a single ground-truth path, allowing for more flexible and realistic performance analysis even with small LLMs.

- **A comprehensive, multi-dimensional assessment methodology** that evaluates an agent based on crucial yet often overlooked metrics (i.e., efficiency, hallucination, and adaptivity) offering a holistic view of the agent's capabilities beyond final accuracy.

- **An in-depth analysis and actionable insights** of current LLM agent performance on complex tool-use tasks, revealing key trends and suggesting concrete strategies for improving their reasoning and reliability.

## 2. Related Works

The capabilities of LLMs have been significantly extended by tool-augmented agents (Zhao et al., 2023), a paradigm built on foundational reasoning techniques like Chain-of-Thought (CoT) (Wei et al., 2022; Kojima et al., 2022) and the ReAct framework (Yao et al., 2023), which enables dynamic planning by interleaving reasoning with tool interactions. This has spurred an explosion of diverse applications, from agents like ToolLLM that master thousands of APIs (Qin et al., 2024) to specialized agents for domains like mathematics and medicine (Gou et al., 2024; Li et al., 2024a), alongside expansion into multi-modal use and research on tool interaction efficiency (Gao et al., 2025; Yuan et al., 2025; Qu et al., 2025).

The rapid development of complex agents necessitates robust evaluation benchmarks, such as GAIA for real-world questions and MetaTool for tool selection (Mialon et al., 2023; Huang et al., 2024). However, a significant limitation of many benchmarks is their reliance on final-answer accuracy and single ground-truth trajectories (e.g., GTA, m&m's), which penalizes valid alternative solutions and scales poorly (Wang et al., 2024; Ma et al., 2024). While this has spurred a shift towards process-level evaluation, even recent benchmarks like ToolBEHonest for hallucination and PIPA for state consistency tend to focus on a single dimension (Zhang et al., 2024b; Kim et al., 2025). A key remaining challenge is that multiple valid trajectories for a single task can lead to score variance, especially with multiple inputs like images (Fig. 1). For complete related works, please refer to the Appendix A.

## 3. Proposed Methodology

In this section, we introduce our proposed framework, TRACE (**T**rajectory-based **R**easoning **A**ssessment and **C**omprehensive **E**valuation). TRACE is a simple yet effective evaluation framework designed to assess the reasoning capabilities of tool-augmented agents operating on the ReAct framework (Yao et al., 2023). Unlike conventional metrics that primarily focus on final answer accuracy, TRACE provides a multi-faceted analysis of an agent's performance by evaluating its entire reasoning trajectory across three critical dimensions: **Efficiency**, **Hallucination**, and **Adaptivity**. A key advantage of our framework is its ability to perform this comprehensive evaluation *without reliance on pre-defined, ground-truth trajectories*, which are often restrictive and expensive to create. The overall framework of TRACE is presented in Fig. 2.

### 3.1. Preliminaries: Formalizing an Agent's Trajectory

A tool-augmented LLM agent interacts with external tools to solve a given user query, $Q$. The entire process of solving $Q$ is captured as a trajectory, $\mathcal{T}$, which is an ordered sequence of steps. We formally define a trajectory as: $\mathcal{T} = (s_1, s_2, ..., s_n)$ where $s_t$ represents the $t$-th step in the reasoning process, and $s_n$ is the final step where the agent produces the final answer. Each individual step $s_t$ for $t \in [1, n-1]$ is a tuple composed of four elements generated sequentially within the ReAct loop $s_t = (th_t, a_t, i_t, o_t)$ where:

- $th_t$: The *thought* generated by the agent. This is a textual rationale where the agent analyzes the current state and decides on the subsequent action.

- $a_t$: The *action* selected by the agent, which corresponds to choosing a specific tool from a predefined set of available tools, $\mathcal{A}$.

- $i_t$: The *action input*, which are the arguments or parameters passed to the selected tool $a_t$.

- $o_t$: The *observation*, which is the output returned by the external tool after executing $a_t$ with input $i_t$. This observation serves as the context for the next step, $s_{t+1}$.

The final step, $s_n$, concludes the trajectory and contains the final answer, $Ans_{final}$, derived from the preceding steps.

### 3.2. The Evidence Bank

The central component of the TRACE framework is the *evidence bank*, denoted as $\mathcal{E}$. The *evidence bank* is a dynamically constructed knowledge base that stores the factual information gathered by the agent throughout its trajectory. It serves as the foundation for evaluating the logical consistency and efficiency of the agent's reasoning process. We find that structurally specifying the relationship between inputs, tools, and their resulting outputs, and storing this information in the *evidence bank* is more effective for measuring efficiency and hallucination than simply providing the full, unstructured dialog to an LLM evaluator.

At each step $t = 1, 2, 3...$, the agent generates a new piece of evidence, $e_t$, which is a tuple of the action taken, the input provided, and the resulting observation: $e_t = (a_t, i_t, o_t)$, and this new evidence is then appended to the *evidence bank*. The state of the *evidence bank* at the end of step $t$, denoted $\mathcal{E}_t$, is the cumulative set of all evidence collected up to that point: $\mathcal{E}_t = \{e_1, e_2, ..., e_t\} = \bigcup_{k=1}^{t}\{e_k\}$. This incrementally built bank provides a complete and objective record of the agent's interaction with its environment, which is crucial for our ground-truth-free evaluation metrics.

### 3.3. Trajectory Evaluation Metrics

TRACE evaluates an agent's trajectory $\mathcal{T}$ using three distinct metrics (i.e., efficiency, hallucination, and adaptivity), each targeting a fundamental aspect of robust reasoning.

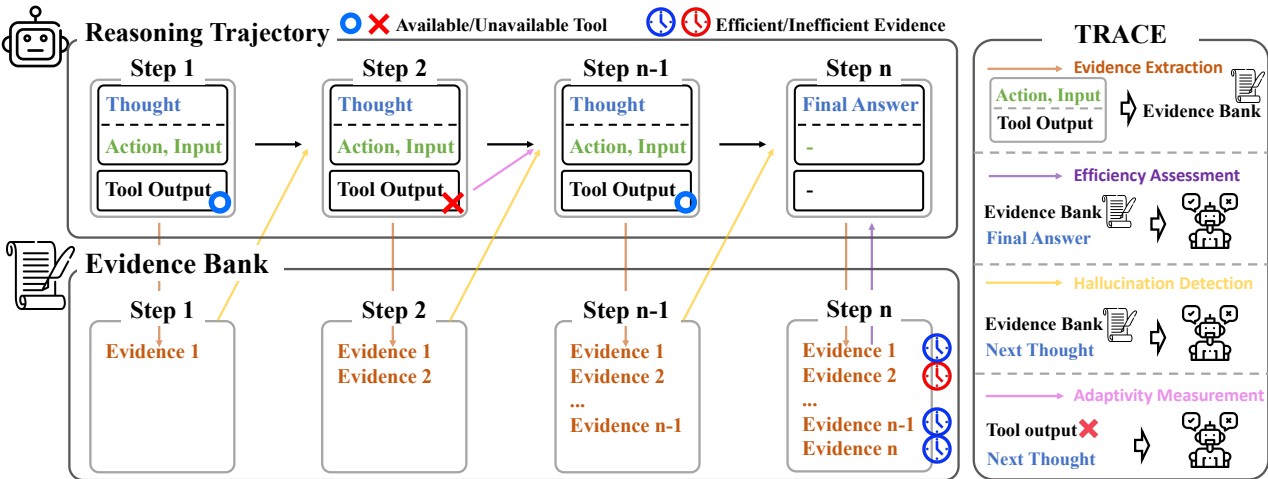

*Figure 2.* Tool outputs are stored in the evidence bank, which is used to detect hallucinations in each thought and to assess trajectory efficiency after the final answer. Adaptivity is measured following the use of an unavailable tool.

**Efficiency.** An ideal agent should reach the correct answer through the most direct (shortest) path possible, without performing inefficient actions. TRACE measures efficiency by quantifying the amount of unnecessary evidence collected in the trajectory. This assessment is performed post-hoc, after the agent has successfully produced the final answer, $Ans_{final}$. To achieve this, we employ an LLM evaluator, tasked with identifying the minimal subset of evidence by examining the final answer $Ans_{final}$ with the complete *evidence bank* $\mathcal{E}_n$. It then selects and retains only the subset of evidence that are essential for logically deducing the answer, and this minimal subset is denoted as $\mathcal{E}_{min} \subseteq \mathcal{E}_n$. The evidence not included in this minimal set is considered unnecessary, so the set of unnecessary evidence is denoted as $\mathcal{E}_{unnecessary} = \mathcal{E} \backslash \mathcal{E}_{min}$. The efficiency score, Eff($\mathcal{T}$), is then calculated as the ratio of the amount of necessary evidence to the total evidence collected:

$$\text{Eff}(\mathcal{T}) = \frac{|\mathcal{E}_{min}|}{|\mathcal{E}_n|} = 1 - \frac{|\mathcal{E}_{unnecessary}|}{|\mathcal{E}_n|}$$

An efficiency score of 1 indicates a perfectly streamlined trajectory with no wasted steps, while a lower score signifies a less efficient reasoning process.

While TRACE does not guarantee the derivation of the optimal solution (i.e., the answer obtained using the minimum number of tools across all possible paths), finding the optimal path without ground-truth (GT) supervision is an extremely hard problem, becoming an intractable task especially as the number of available tools increases. Consequently, instead of attempting to solve this intractable optimization problem, TRACE focuses on measuring efficiency by detecting unnecessary tools within the set of tools the agent chose to use.

**Hallucination.** Hallucination in tool-augmented agents occurs when the agent's internal thought process deviates from the established facts. TRACE identifies hallucinations by assessing whether an agent's thought at a given step is logically derivable from the evidence collected so far.

Specifically, for each step $t$, we evaluate the thought $th_t$. A thought is considered a hallucination if it contains information or makes assumptions that cannot be substantiated by the contents of the *evidence bank* from the previous steps, $\mathcal{E}_{t-1}$. We define a boolean validation function, IsGrounded($th_t, \mathcal{E}_{t-1}$), which is instantiated using the LLM evaluator. A hallucination is detected if this function returns false. The prediction of hallucination for a single step, $H(s_t)$, is defined as:

$$H(s_t) = \begin{cases} 1 & if \, \neg \, \text{IsGrounded}(th_t, \mathcal{E}_{t-1}) \\ 0 & otherwise \end{cases}$$

For the first step, the *evidence bank* $\mathcal{E}_0$ is emtpy, so the model detect the hallucination of the agent only depend on its thought $th_1$. The hallucinations score in a trajectory $\mathcal{T}$ is the average of hallucination counts for each step: $H(\mathcal{T}) = \sum_{t=1}^{n} H(s_t)/n$.

In this paper, we define hallucination as occurring even when the model uses internal knowledge not present in the evidence bank to determine whether a tool-augmented LLM agent can select and utilize the appropriate tool for a given question following recent study (Li et al., 2025).

**Adaptivity.** Real-world scenarios are imperfect; tools can fail due to API changes, version incompatibilities, or other issues such as timeout error and rate-limiting error, and a robust agent should be able to adapt to such failures. TRACE measures adaptivity by evaluating the agent's response to an unavailable tool, when an observation $o_t$ in-

*Table 1.* Performance of a naive approach for evaluating reasoning trajectories (i.e., LLM-as-a-Judge) and TRACE on our meta-evaluation datasets, **Meta-GTA** and **Meta-m&m's**.

| Models | Meta-GTA | | | | | | Meta-m&m's | |
|---|---|---|---|---|---|---|---|---|
| | LLM-as-a-Judge | | | TRACE | | | LLM-as-a-Judge | TRACE |
| | Efficiency | Hallucination | Adaptivity | Efficiency | Hallucination | Adaptivity | Efficiency | Efficiency |
| Claude-Sonnet-4 | 86.08 | 89.68 | 98.83 | 94.64 (+8.56) | 95.21 (+5.53) | 99.63 (+0.8) | 86.08 | 85.75 (-0.33) |
| GPT-4.1 | 81.45 | 94.42 | 97.95 | 94.24 (+12.79) | 95.40 (+0.98) | 97.03 (-0.92) | 84.35 | 86.12 (+1.77) |
| o3-mini | 90.24 | 94.68 | 96.59 | 94.09 (+3.85) | 94.69 (+0.01) | 96.91 (+0.32) | 88.13 | 88.56 (+0.43) |
| Llama-3.3-70B | 76.55 | 88.95 | 98.23 | 90.03 (+13.48) | 95.97 (+7.02) | 98.30 (+0.07) | 86.47 | 87.19 (+0.72) |
| Llama-3.1-8B | 55.67 | 89.59 | 78.98 | 70.46 (+14.79) | 93.78 (+4.19) | 85.28 (+6.3) | 44.61 | 64.05 (+19.44) |

dicates a tool execution failure (e.g., returns an API error). Let $\mathcal{F}$ be the set of step indices where such failures occur, then we assess the subsequent step $s_{t+1}$ for each $t \in \mathcal{F}$. The adaptivity score for a failure event at step $t$, $Adp(s_t)$, is a binary value (1 for adaptive, 0 for non-adaptive) determined by an LLM evaluator. The agent is considered adaptive, $Adp(s_{t+1}) = 1$, if its thought of the subsequent thought $th_{t+1}$ acknowledges the failure and its action $a_{t+1}$ represents a sensible alternative strategy. Otherwise, it is predicted as non-adaptive, $Adp(s_{t+1}) = 0$. For instance, an adaptive agent might select a different tool with similar functionality or modify its approach to proceed without the failed tool, while a non-adaptive agent repeatedly tries the same failed tool or becomes stuck. The purpose of this metric metric is to assess whether the agent can utilize a different tool, regardless of the issue encountered, so the adaptivity metric remains valid and applicable to various issues.

## 4. Experiments: Meta-evaluation of TRACE

### 4.1. Meta-GTA and Meta-m&m's: Meta-Evaluation dataset

To verify that TRACE accurately evaluates the trajectories returned by agents, we introduce a dataset for meta-evaluation, termed as such because its purpose is to evaluate the evaluation framework (TRACE) itself. We validate the effectiveness of TRACE by augmenting and labeling the latest multimodal tool-augmented agent benchmark datasets, GTA (Wang et al., 2024), and m&m's (Ma et al., 2024). GTA contains a small number of high-quality, human-curated samples, each including a single ground-truth trajectory with the actions, action inputs, and thoughts that an LLM should return. Based on GTA, we generate **Meta-GTA** by augmenting each ground-truth trajectory into multiple valid ground-truth trajectories, carefully considering the dependencies between tool orders. On top of these trajectories, we then strategically insert inefficient steps, hallucinatory thoughts, and adaptive actions following the selection of an unavailable tool, and annotate all steps with corresponding labels. To construct the **Meta-m&m's** dataset, we first

augment each ground-truth trajectory into a set of multiple valid ground-truth trajectories. After augmentation, we deliberately insert inefficiency steps and annotate them with the corresponding labels. Because the m&m's dataset is structured for LLMs to generate all necessary actions at once without intermediate reasoning, we exclude hallucination and adaptivity steps A more detailed description of the dataset including creation and validation process is presented in Appendix B.

For the **Meta-GTA**, the accuracy of TRACE for hallucination ($Acc_H$), efficiency ($Acc_{Eff}$), and adaptivity ($Acc_{adp}$) is calculated as follows, using the labels in the data, $H_{label}(s_t)$, $I_{label}(s_t)$, and $Adp_{label}(s_t)$, respectively.

$$Acc_H(\mathcal{T}) = \frac{1}{n} \sum_{t=1}^{n} \mathbb{I}(H(s_t) = H_{label}(s_t)),$$

$$Acc_{Eff}(\mathcal{T}) = \frac{1}{n} \sum_{t=1}^{n} \mathbb{I}(I(s_t) = I_{label}(s_t))$$

$$Acc_{adp}(\mathcal{T}) = \frac{1}{|\mathcal{F}|} \sum_{t \in \mathcal{F}} \mathbb{I}(Adp(s_{t+1}) = Adp_{label}(s_{t+1}))$$

Due to the structure of the **Meta-m&m's** dataset, where ground-truth labels consist of a complete sequence of actions without intermediate thought or reasoning, the efficiency accuracy for this dataset was determined as a binary score. A score of 1 was awarded if the predicted count of inefficient steps exactly matched the label; otherwise, the score was 0.

### 4.2. Meta-Evaluation Results

The results of TRACE's evaluation performance on our generated dataset are presented in Table 1. We conduct the experiments with various proprietary LLMs and open-source models of different sizes, also including reasoning models. Detailed description about experimental setting is available in Appendix B.3.

We find that most LLM models can effectively evaluate the efficiency, hallucination, and adaptivity of tool-augmented

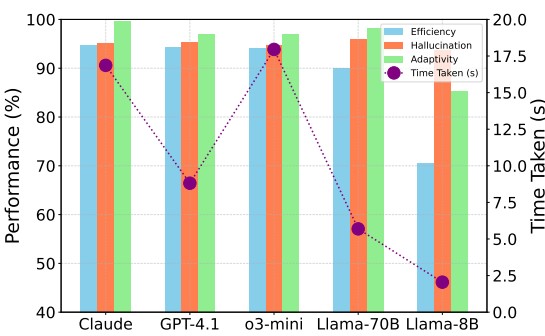

*Figure 3.* Time Efficiency Comparison of LLM Evaluators using TRACE on **Meta-GTA** dataset.

*Table 2.* Comparison between TRACE and PIPA for cases where multiple trajectories exist for the same query (**Meta-GTA**).

| Models | TRACE | PIPA |
| --- | --- | --- |
| Claude-Sonnet-4 | 98.68 ± 1.26 | 77.31 ± 9.96 |
| GPT-4.1 | 96.91 ± 2.28 | 84.04 ± 7.17 |
| o3-mini | 95.48 ± 1.81 | 89.76 ± 4.74 |
| Llama-3.3-70B | 94.32 ± 4.40 | 83.28 ± 8.57 |
| Llama-3.1-8B | 79.10 ± 8.73 | 24.94 ± 12.91 |

*Table 3.* Comparison between TRACE and PIPA for all trajectories in **Meta-GTA**.

| Models | TRACE | PIPA |
| --- | --- | --- |
| Claude-Sonnet-4 | 94.64 | 84.46 |
| GPT-4.1 | 94.24 | 81.61 |
| o3-mini | 94.09 | 80.06 |
| Llama-3.3-70B | 90.03 | 88.11 |
| Llama-3.1-8B | 70.46 | 41.20 |

agents without relying on ground-truth trajectories when using TRACE. Notably, this demonstrates that evaluation is sufficiently achievable with open-source models, avoiding the evaluation costs associated with large proprietary models. It is worth noting that the dataset was constructed by injecting unnecessary steps into efficient trajectories and then labeling them for TRACE to detect. If TRACE accurately identifies the inserted inefficient step, the remaining steps constitute the clean, efficient trajectory, and the accuracy metric is specifically designed to reward the identification of this intentionally inserted inefficient step. This effectiveness is attributed to TRACE's framework, which utilizes the *evidence bank*. To validate the benefit of the evidence bank, we further test the LLM-as-a-Judge method that evaluates entire trajectories without the evidence bank. In Table 1, we observe that TRACE achieves more accurate assessments across efficiency, hallucination, and adaptivity. Moreover, we observe performance improvements across all LLMs when using TRACE, the most significant gains were observed in the smaller open-source models. This result validates the effectiveness of the *evidence bank*, demonstrating that it enables accurate evaluation even with smaller, more efficient models.

We also measure the average time required per query for each model to evaluate **Meta-GTA** to verify how efficiently we can evaluate agents using TRACE. Figure 3 highlights the efficiency advantage of Llama-3.3-70B. Although it performs on par with larger proprietary models (Table 1), its average evaluation time is reduced by as much as a factor of three. This indicates that TRACE not only enables accurate evaluation but also facilitates efficient assessment in terms of cost and time with smaller models, which becomes more significant when evaluating large data.

## 4.3. Comparison with An Existing Trajectory Evaluation Method

To validate the effectiveness of calculating efficiency through the evidence bank, we compare TRACE with an existing trajectory evaluation method. Because existing approaches do not directly assess efficiency, hallucination, and adaptivity, we adopt a baseline a method that evaluates trajectories from related perspectives. Specifically, PIPA (Kim et al., 2025) proposes a metric called *state consistency*, to measure the consistency between the previous and current state. Since the consistency score penalizes the use of unnecessary tools in executing a user's query, we expect it to correlate with our *efficiency* metric.

**Robustness to multiple trajectories with multi inputs.** Table 2 presents the robustness of evaluation for the efficiency of TRACE and the *state consistency* measured by PIPA. To verify whether our evaluation is robust even in the case where there exist multiple valid trajectories for solving a single task, we sample only those from the Meta-GTA dataset and calculate their efficiency, along with the standard deviation among them. Note that all of corresponding trajectories for a query are correct and efficient; they differ only in the sequence of tool calls. In this experiment, therefore, an ideal outcome is the efficiency score close to 100 with a standard deviation is close to 0.

The results show that our proposed method measures efficiency more accurately while also having a lower standard deviation, demonstrating that our method consistently evaluate efficiency regardless of the specific trajectory taken. In the case of PIPA, which measures the consistency of the current state based on accumulated states, we observe that the consistency score decreases when the agent has to handle multiple input files provided with the query in parallel. In contrast, TRACE identifies evidence within the entire trajectory that is unnecessary for reaching the final answer. Consequently, the diversity among trajectories does not affect TRACE's performance.

**Effectiveness of efficiency evaluation.** We also provide the performance of evaluating trajectory efficiency in Table 3.

Since *state consistency* is similar to our efficiency metric, we can confirm that PIPA is also capable of detecting inefficient steps, but TRACE evaluates efficiency of the trajectory more precisely across all models. Notably, in the case of smaller models like Llama-8B, its accuracy drops sharply compared to TRACE. This result once again proves that the *evidence bank*, which is a core mechanism of our proposed method, enables the effective measurement of efficiency while PIPA relies on unrefined full dialog.

# 5. Evaluation of Tool-augmented LLM Agents with TRACE

While existing research of tool-augmented benchmarks like the GTA dataset (Wang et al., 2024) have predominantly focused on final answer accuracy, this single metric can obscure crucial differences in the quality of the underlying reasoning. In this section, we go beyond this limitation by conducting experiments to evaluate not only the answer accuracy of various LLM agents but also the quality of their reasoning trajectories across three dimensions: efficiency, hallucination, and adaptivity. We aim to demonstrate that even agents with seemingly comparable accuracy can exhibit performance disparities when their reasoning processes are closely examined. Through this deeper analysis, we seek to identify common causes of agent failure and highlight the distinct characteristics of each model.

## 5.1. Experimental Settings

Our experimental setup involves diverse LLMs to serve as agents. This includes proprietary models and a range of open-source models of varying sizes, as well as reasoning model. The agents under evaluation are **1) Claude-Sonnet**, **2) GPT-4.1**, **3) o3-mini**, **4) Llama 70B**, **5) Llama 8B**, **6) Mixtral 8x7B**, **7) Mistral 7B**, **8) Qwen 72B**, and **9) Qwen 7B**, where the first three models are proprietary models. We task each of these agents with solving problems from the GTA (Wang et al., 2024) dataset, a recent and challenging benchmark featuring multimodal and multi-input queries.

We employ our TRACE framework to conduct a fine-grained evaluation of each agent's reasoning trajectory. For the evaluator models within TRACE, we select **Claude-Sonnet**, **GPT-4.1**, **Llama 70B**, and **o3-mini**, based on their strong performance in our meta-evaluation as shown in Table 1. These evaluators assess each agent's trajectory across the three core dimensions: efficiency score, hallucination reduction score, and adaptivity score, where the efficiency of a trajectory is measured only on the condition that it produced the correct answer. To rigorously evaluate adaptivity, we augment the standard toolset in GTA, incorporating a set of fake tools with names syntactically similar to the original, valid tools. When an agent selects one of these fake tools, the environment returns an "unavailable tool" error message.

This controlled-failure setup allows us to systematically observe whether the agent can flexibly select an alternative tool in its subsequent step.

Beyond our TRACE framework, we conduct a more detailed analysis by incorporating additional metrics for instruction following and answer accuracy, adhering to the evaluation protocol established by the GTA benchmark (Wang et al., 2024). We measure **Instruction Error**, which is divided into two categories: a non-existent tool selection and failing to provide arguments in the correct format for a chosen tool. We count each error and present its ratio relative to the total length of the trajectory. We also present **Answer Accuracy** depending on the query type. For multiple-choice questions (MCQ), we use binary accuracy, while we use cosine similarity of embeddings from All-MPNet-Base-V2 (Song et al., 2020) for queries requiring a long-form textual response (LTR). For queries where the answer is an image (IMG), we measure the cosine similarity of arguments embeddings following the protocol of GTA (Wang et al., 2024). We also report an **Overall Accuracy**, calculated as the micro-average across all individual samples (MCQ, LTR, and IMG) to accurately reflect the distribution of query types in the dataset.

**Feedback Mechanism.** To ensure that trajectories are not prematurely terminated due to minor, recoverable errors, we implement a feedback mechanism. If an agent attempts to call a tool that is not in the provided toolset or uses an incorrect argument format, the environment provides specific feedback (i.e., "tool not in the list" or "tool execution error") instead of halting the process. This approach allows us to gather more comprehensive trajectories for analysis, preventing trivial mistakes from obscuring an agent's broader reasoning capabilities. It is crucial to note that while this feedback allows the agent to proceed, the initial errors are still logged and counted towards the "non-existent tool selection" and "wrong argument format" metrics, respectively. More detailed experimental settings are available in Appendix C.

**Extension to Multi-Agent System.** Thanks to its design as an widely applicable framework, TRACE can evaluate the trajectories of more sophisticated agents such as Multi-Agent Systems (MAS) in addition to standard ReAct-style agentic systems. TRACE enables assessment of both individual agent performance and overall system performance by constructing an evidence bank for each agent, verifying hallucinations in both the orchestrator and the agents at every step using all available evidence banks, and examining whether unnecessary evidence was used once the correct final answer is reached. To demonstrate the extensibility of TRACE across diverse datasets and agent types, we additionally showcase the efficiency and adaptivity of trajectories produced by the multi-agent system Magentic-

*Table 4.* Performance of LLM agents on GTA dataset. **Avg.** denotes average of performance from all evaluators, and **Inst.** denotes Instruction Error (non-existent tool selection / wrong argument format).

| Model | Claude-Sonnet-4 | | | | | GPT-4.1 | | | | | o3-mini | | | | |
|---|---|---|---|---|---|---|---|---|---|---|---|---|---|---|---|
| Evaluator | Claude | GPT | Llama | o3-mini | Avg. | Claude | GPT | Llama | o3-mini | Avg. | Claude | GPT | Llama | o3-mini | Avg. |
| Efficiency | 0.9427 | 0.9495 | 0.9459 | 0.9025 | **0.9351** | 0.9659 | 0.9328 | 0.9380 | 0.8958 | **0.9331** | 0.9888 | 0.9457 | 0.9269 | 0.8914 | **0.9382** |
| Hallucination | 0.9511 | 0.9815 | 0.9878 | 0.9718 | **0.9730** | 0.9513 | 0.9953 | 0.9884 | 0.9787 | **0.9784** | 0.9818 | 0.9961 | 0.9961 | 1.0000 | **0.9935** |
| Adaptivity | 0.9091 | 0.9545 | 0.8636 | 0.7727 | **0.8750** | 0.7667 | 0.8111 | 0.8333 | 0.8111 | **0.8056** | 0.5909 | 0.6818 | 0.6818 | 0.5455 | **0.6250** |
| Inst. ↓ | 0.0029 / 0.0038 | | | | | 0.0074 / 0.0093 | | | | | 0.0083 / 0.0400 | | | | |
| Answer Accuracy | 0.5321 / 0.6607 / 0.6754 | | | | | 0.4487 / 0.7208 / 0.6914 | | | | | 0.4808 / 0.7320 / 0.5933 | | | | |
| Overall Accuracy | 0.5767 | | | | | 0.5281 | | | | | 0.5263 | | | | |
| Model | Llama-3.3-70B | | | | | Mixtral-8x7B | | | | | Qwen-72B | | | | |
| Evaluator | Claude | GPT | Llama | o3-mini | Avg. | Claude | GPT | Llama | o3-mini | Avg. | Claude | GPT | Llama | o3-mini | Avg. |
| Efficiency | 0.8784 | 0.7456 | 0.7819 | 0.7064 | **0.7781** | 0.7513 | 0.7513 | 0.7811 | 0.6858 | **0.7424** | 0.9687 | 0.9245 | 0.9287 | 0.9525 | **0.9436** |
| Hallucination | 0.8090 | 0.9410 | 0.9768 | 0.9214 | **0.9121** | 0.8616 | 0.9505 | 0.9831 | 0.9381 | **0.9333** | 0.9324 | 0.9820 | 0.9836 | 0.9052 | **0.9508** |
| Adaptivity | 0.8547 | 0.9012 | 0.9012 | 0.8721 | **0.8823** | 0.5000 | 0.6250 | 0.5000 | 0.6250 | **0.5625** | 0.7969 | 0.7969 | 0.7969 | 0.7969 | **0.7969** |
| Inst. ↓ | 0.0738 / 0.0047 | | | | | 0.091 / 0.061 | | | | | 0.017 / 0.0021 | | | | |
| Answer Accuracy | 0.3205 / 0.3752 / 0.5191 | | | | | 0.0109 / 0.6223 / 0.1822 | | | | | 0.4359 / 0.7679 / 0.6815 | | | | |
| Overall Accuracy | 0.3738 | | | | | 0.1631 | | | | | 0.5202 | | | | |
| Model | Llama-3.1-8B | | | | | Mistral-7B | | | | | Qwen-7B | | | | |
| Evaluator | Claude | GPT | Llama | o3-mini | Avg. | Claude | GPT | Llama | o3-mini | Avg. | Claude | GPT | Llama | o3-mini | Avg. |
| Efficiency | 0.5244 | 0.7423 | 0.5184 | 0.4633 | **0.5621** | - | - | - | - | **-** | 0.8893 | 0.9445 | 0.9118 | 0.8650 | **0.9026** |
| Hallucination | 0.6823 | 0.8355 | 0.9519 | 0.8069 | **0.8192** | 0.9959 | 1.0000 | 1.0000 | 1.0000 | **0.9990** | 0.8282 | 0.9435 | 0.9566 | 0.9193 | **0.9119** |
| Adaptivity | 0.5556 | 0.5556 | 0.5556 | 0.5556 | **0.5556** | 0.0000 | 0.0000 | 0.0000 | 0.0000 | **0.0000** | 0.8495 | 0.8656 | 0.8656 | 0.8979 | **0.8696** |
| Inst. ↓ | 0.2397 / 0.0027 | | | | | 0.0480 / 0.061 | | | | | 0.1254 / 0.008 | | | | |
| Answer Accuracy | 0.0321 / 0.2699 / 0.6193 | | | | | 0.0000 / 0.1890 / 0.0088 | | | | | 0.2628 / 0.5944 / 0.6823 | | | | |
| Overall Accuracy | 0.1948 | | | | | 0.0154 | | | | | 0.3904 | | | | |

One (Fourney et al., 2024) on the state-of-the-art benchmark dataset GAIA (Mialon et al., 2023), as presented in Appendix D.1.

## 5.2. Experiment Results

Comprehensive results of our experiments are summarized in Table 4. Here, we provide a detailed analysis of these findings, focusing on overall performance, the three core TRACE metrics, and instruction following capabilities.

### 5.2.1. OVERALL PERFORMANCE ANALYSIS

In terms of overall accuracy, proprietary models demonstrate the strongest performance as expected. However, their results in these specialized tool-augmented tasks suggest considerable room for improvement when compared to their powerful general-purpose capabilities.

Among the open-source LLMs, a clear trend emerges within the same model family: larger models consistently outperform their smaller counterparts. However, across the entire open-source landscape, we observe that model size is not the sole determinant of success. Notably, the Qwen-7B model achieves remarkable performance, surpassing both Llama-3.3-70B and Mixtral-8x7B. Furthermore, Qwen-72B performs at a level comparable to proprietary models like Claude and GPT. This finding strongly implies that there are specific architectural features or helpful training methodolo-

gies to create effective tool-augmented agents. We believe this presents a valuable future direction for research aimed at developing more capable agents.

Furthermore, powerful proprietary models have a very low instruction error rate, whereas open-source models, particularly smaller ones, exhibit higher error rates. This is likely because smaller models face greater difficulty in processing long-context instructions accurately. This implies that smaller models, when deployed as tool-augmented agents, would significantly benefit from an accompanying correction mechanism that validates tool selection and argument formatting.

### 5.2.2. ANALYSIS ON THE THREE TRACE METRICS

**Efficiency.** We measure efficiency only for successful trajectories, that is, only when they lead to the correct final answer. The efficiency results largely follow the trends observed in overall accuracy. While larger models tend to construct more efficient trajectories, this is not always the case. A key insight from this analysis is that as an agent generates a more inefficient (i.e., longer) trajectory, its likelihood of producing an incorrect answer increases. We hypothesize this is because longer trajectories require the agent to process a longer context, which can weaken its ability to perform accurate reasoning. It is worth noting that Mistral-7B has no efficiency score, as it fails to produce any correct answers in our experiments. Further analysis of the efficiency of trajec-

tories where the model failed to derive the correct answer is available in Appendix D.2

**Hallucination.** The hallucination metric also shows a high correlation with overall accuracy, as a higher hallucination naturally reduces the probability of reaching a correct answer. Notably, the reasoning model, o3-mini, demonstrates a remarkably low hallucination rate. This is likely attributable to the inherent nature of Large Reasoning Models (LRMs) to perform deep reasoning (Plaat et al., 2024; Chen et al., 2025), focusing strictly on the accumulated evidence. In the case of Mistral-7B, we observe that it achieves a high score (i.e., a low rate of hallucination) not by successfully completing tasks, but by terminating its operation when it determines it cannot proceed.

**Adaptivity.** Adaptivity is another metric directly linked to overall accuracy. To measure this, we test whether an agent can select a valid tool after an initial attempt to use a pre-defined, unavailable tool with a similar name. We find that GPT-4.1 and Qwen-72B, despite their high overall accuracy, exhibit a relatively low adaptivity score. This suggests that enhancing the ability to adapt to tool failures could be a key factor in further elevating their performance.

In summary, leveraging TRACE enables a more in-depth evaluation of agents than was previously possible. For instance, if we only measure the Overall Score, as is common in other studies, the Qwen-72B, GPT-4.1, and o3-mini models would appear to have similar accuracy and thus be considered interchangeable. However, our results reveal that Qwen is more efficient than GPT-4.1 but shows higher hallucination rate. On the other hand, despite having almost no hallucinations, o3-mini struggles to continue its reasoning robustly after a tool-calling failure (low adaptability). Similarly, Llama-70B and Qwen-7B show comparable overall scores, yet they differ significantly in efficiency and adaptivity. A similar pattern is observed between Llama-8B and Mixtral-8x7B, which have similar overall scores but large gaps in efficiency and hallucination. This implies that the strategies required to enhance the performance of each model on tool-augmented tasks are distinct. Furthermore, it empowers users to select a model based on their specific priorities when building an agent; whether to prioritize higher efficiency, greater reliability, or more robust reasoning capabilities. For specific examples illustrating these points, please refer to the case studies in Appendix E.

### 5.2.3. QUANTITATIVE ANALYSIS OF TOKEN CONSUMPTION

In this section, we analyze token usage as a potential cause of performance degradation in tool-augmented agents. Figure 4 illustrates the overall accuracy, the average number of turns per query, where a turn corresponds to a reasoning step involving a tool call, and the number of output tokens

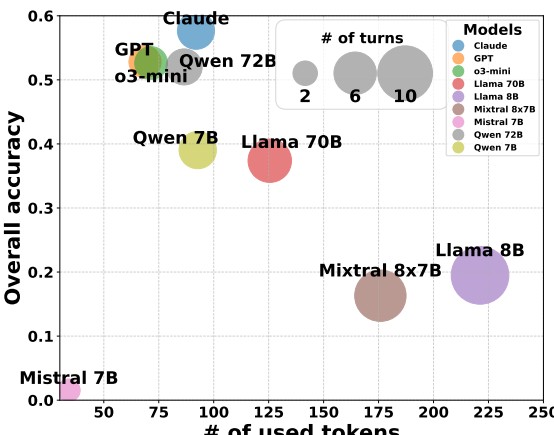

*Figure 4.* Model accuracy based on the number of tokens used and dialogue turns.

per query for each model on the GTA dataset. (Note: to standardize the output token count across all models, we exclude the reasoning tokens for o3-mini). With the exception of Mistral 7B, which produced few meaningful answers, we observe a general trend that models using more turns tend to have lower performance. This is because they often create inefficient trajectories with unnecessary steps, ultimately failing to arrive at the correct answer. Notably, we can also see a clear negative correlation between the number of output tokens and the overall accuracy. Exceeding the minimum number of tokens required for reasoning can introduce unnecessary information or noise, which directly degrades performance. This creates a longer context, a problem that becomes particularly significant for smaller models. This result suggests that for smaller, low-confidence models, limiting the token count, rather than allowing them to think more, can actually improve the performance of tool-augmented agents.

## 6. Conclusion

In this paper, we address the limitations of conventional evaluation methods for tool-augmented LLM agents, which predominantly focus on final answer accuracy while overlooking the crucial reasoning process. To overcome this, we introduced TRACE, a simple yet effective framework for the comprehensive evaluation of an agent's reasoning trajectory. By operating without reliance on a single, predefined ground-truth path, TRACE offers a more realistic and scalable assessment across three critical dimensions: efficiency, hallucination, and adaptivity. Using TRACE allows for a more accurate understanding of the performance of tool-augmented agents, which helps in developing more effective agents.

## Impact Statement

This paper presents work whose goal is to advance the field of Machine Learning. There are many potential societal consequences of our work, none which we feel must be specifically highlighted here.

## Reproducibility Statement

The complete source code for our proposed framework is available at https://github.com/wonjoong-kim/TRACE. Proposed dataset used for meta-evaluation is provided in the supplementary materials. A detailed description of the construction, augmentation, and labeling process for our meta-evaluation dataset including the used prompts can be found in Appendix B.1, and Appendix B.3 provides a comprehensive list of all hyperparameter settings and the computational infrastructure used to obtain our results of meta-evaluation. Furthermore, we present detailed description of experimental settings in Appendix C, ensuring that our experimental findings can be precisely replicated.

## Acknowledgements

This work was supported by Institute of Information & Communications Technology Planning & Evaluation(IITP) grant funded by the Korea government(MSIT) (RS-2023-00216011), the National Research Foundation of Korea(NRF) grant funded by the Korea government(MSIT) (RS-2024-00406985), and National Research Foundation of Korea(NRF) funded by Ministry of Science and ICT (RS-2022-NR068758).

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

# A. Complete Related Work

## A.1. Tool-augmented LLM Agent

The capabilities of LLMs have been significantly extended by integrating external tools, giving rise to tool-augmented agents (Zhao et al., 2023). Foundational to this paradigm is the ability of LLMs to generate intermediate reasoning steps. Early work on Chain-of-Thought (CoT) prompting demonstrated that eliciting a series of intermediate steps could improve performance on complex reasoning tasks (Wei et al., 2022; Kojima et al., 2022). Building directly on this, the ReAct framework (Yao et al., 2023) established a synergistic model where LLMs interleave reasoning traces ("thought") with external tool interactions ("action"). This enables agents to plan, execute, and adjust their strategies to solve tasks dynamically.

Following these seminal works, the field has seen an explosion of agents designed for diverse and complex applications. Some research has focused on enabling agents to master a vast array of tools. For instance, ToolLLM (Qin et al., 2024) demonstrated the ability to facilitate LLMs in using over 16,000 real-world APIs, showcasing remarkable generalization in tool use. Others have focused on creating specialized agents for specific domains that demand high precision, such as mathematical problem-solving with TORA (Gou et al., 2024), medical task assistance with MMedAgent (Li et al., 2024a), and financial trading with multimodal foundation agents (Zhang et al., 2024a). Concurrently, the scope of tool use has expanded into the multi-modal domain, with models like MLLM-Tool (Wang et al., 2025) and other vision-language model-driven agents that can interpret and act upon visual information (Gao et al., 2025). As complexity has grown, efforts have also been made to improve the efficiency of tool interaction itself, through methods like providing concise tool instructions (Yuan et al., 2025) or enabling models to self-improve tool documentation (Qu et al., 2025).

## A.2. Evaluation of Tool-Augmented Agents

The rapid development of complex, multi-step agents necessitates robust and comprehensive evaluation benchmarks. A number of benchmarks have been proposed to assess agent capabilities across different tasks. For example, GAIA (Mialon et al., 2023) was designed to test general AI assistants on challenging real-world questions, while MetaTool (Huang et al., 2024) specifically focuses on the agent's ability to decide whether to use a tool and which tool to select from a given set.

However, a significant limitation of many existing benchmarks is *their reliance on the accuracy of the final answer as the primary metric*. Benchmarks such as GTA (Wang et al., 2024) and m&m's (Ma et al., 2024), for instance, are constrained by primarily validating an agent's trajectory against a single, pre-defined ground-truth sequence. This approach not only penalizes agents that discover alternative, valid solution paths but also scales poorly as tool complexity increases, as manually enumerating all possible correct paths is computationally infeasible.

Recognizing this insufficiency, recent work has begun to explore process-level evaluation. While this is a critical step forward, even notable diagnostic benchmarks tend to focus on a single dimension of agent behavior. For instance, ToolBEHonest (Zhang et al., 2024b) measures hallucination, and PIPA (Kim et al., 2025) proposes a more unified protocol to assess behaviors such as state consistency. A holistic evaluation that assesses a broader set of attributes defining a high-quality reasoning process—such as efficiency and adaptivity—remains a significant gap. A remaining challenge is that for a single task, multiple valid trajectories can exist, leading to score variance even among solutions of the same quality, especially with multiple inputs like images (Fig. 1).

# B. Detailed Description of Meta-Evaluation Process

## B.1. Dataset Generation and Validation

In this section, we provide a detailed description of the generation and validation process for **Meta-GTA** and **Meta-m&m's**. For each dataset, we first augment the single ground-truth trajectory into multiple valid ground-truth trajectories for the given query. This augmentation is performed using the GPT-4o model, with the input prompt shown in Fig. 7. Following this, we introduce new inefficient, hallucinatory, and adaptive steps, using the input prompts detailed in Fig. 8, 9, and 10. After inserting each step, we annotate it with the corresponding label to allow for the verification of TRACE's predictions. The original ground-truth trajectory is preserved throughout this process to ensure that the performance evaluation also reflects any incorrect predictions by TRACE, such as misclassifying a normal step as inefficient or a hallucination.

To ensure the integrity and quality of the dataset, a rigorous validation process was established. This process began with generating validation outputs for each data point using three distinct models: Claude Sonnet 4.0, GPT-4o, and Gemini Pro,

employing prompts in Fig. 11, 12,13, and 14 respectively. Following generation, a random sample of 100 outputs from each model was selected for manual inspection. A human annotator then assessed the validity of these samples, achieving 98.79% agreement. The final meta-evaluation dataset is exclusively composed of data points that received a "valid" consensus from all three LLMs.

*Table 5.* Exact version of each model used in Meta-evaluation dataset generation and validation in Section 4, baselines in Section 5, and evaluators for TRACE in Section 4 and 5.

| Model Name | Used Version |
|---|---|
| **OpenAI API** | |
| GPT-5-mini | `gpt-5-mini-2025-08-07` |
| GPT-4.1 | `gpt-4.1-2025-04-14` |
| o3-mini | `o3-mini-2025-01-31` |
| GPT-4o | `gpt-4o-2024-11-20` |
| **Claude API** | |
| Claude-sonnet-4 | `claude-sonnet-4-20250514` |
| **Gemini API** | |
| Gemini-2.5-pro | `gemini-2.5-pro-preview-06-05` |
| **TogetherAI API** | |
| Llama-3.1-8B-Instruct | `meta-llama/Meta-Llama-3.1-8B-Instruct-Turbo` |
| Llama-3.3-70B-Instruct | `meta-llama/Llama-3.3-70B-Instruct-Turbo` |
| Mistral-7B-Instruct | `mistralai/Mistral-7B-Instruct-v0.1` |
| Mixtral-8x7B-Instruct | `mistralai/Mixtral-8x7B-Instruct-v0.1` |
| Qwen2.5-7B-Instruct | `Qwen/Qwen2.5-7B-Instruct-Turbo` |
| Qwen2.5-72B-Instruct | `Qwen/Qwen2.5-72B-Instruct-Turbo` |

## B.2. Statistics of the Dataset

In this section, we provide a detailed statists of our newly constructed meta-evaluation datasets, **Meta-GTA** and **Meta-m&m's**. To offer a clear and comprehensive context for our experimental results, we present a summary of their key statistics in Table 6. This table shows the distribution of the synthetically injected labels, alongside with the number of correct and efficient trajectory. For **Meta-GTA** dataset, from the existing set of 33 tools, we select four (Calculator, OCR, ImageDescription, and GoogleSearch) and add corresponding synthetic tools with similar names (FastCalculator, FastOCR, ImageDescriptor, and WebSearch) to the tool set. All tool descriptions for these synthetic tools are kept identical to the originals, differing only by name. If an agent selects a synthetic tool, the tool output notifies that it is unavailable, at which point we measure the agent's adaptivity by assessing whether it successfully continues its reasoning in the subsequent step.

For the **Meta-m&m's** dataset, we use the `test-human-verified-filtered` split in the dataset, which was human-filtered by the original authors to ensure high data quality. Since the trajectories in this dataset do not contain the agent's thoughts, we limit our augmentation process to only multiple valid path augmentation and the injection of inefficient steps.

*Table 6.* Statistics of **Meta-GTA** and **Meta-m&m's**

| | Meta-GTA | Meta-m&m's |
|---|---|---|
| Total | 761 | 735 |
| Correct | 168 | 374 |
| Inefficient | 171 | 361 |
| Hallucination | 251 | - |
| Adaptivity | 171 | - |
| Unavailable tools | 4 | - |

## B.3. Detailed Experimental Setting of Meta-Evaluation

For the meta-evaluation and subsequent experiments within the TRACE framework, we adopted five models including three proprietary models, where one of them is reasoning model, and two open-source models with different sizes: GPT-4.1,

Claude-Sonnet-4, Llama-3.3-70B, Llama-3.1-8B, and o3-mini. To enhance reproducibility of our experiments, we specify To provide detailed information about the exact used model versions used in experiments in Table 5. Furthermore, we illustrate the input prompts for models to conduct the evaluation of TRACE in Fig. 15, 16, and 17.

## C. Detailed Setting of Experiments

We utilize various LLMs including proprietary models, reasoning model, and open-source model in various sizes in this paper. To provide detailed information about the models used and enhance reproducibility, we specify the exact model versions used in experiments in Table 5.

### C.1. Hyperparameter Setting

To ensure experimental reproducibility, we set the temperature to 0 and fixed the max tokens at 4096 for all trials. Additionally, the maximum number of action turns per query was limited to 10; any query exceeding this limit was automatically treated as a failure.

### C.2. Generation Details

Building upon a prior study (Wang et al., 2024), we developed a tool-agent system utilizing a ReAct-style prompt. However, to improve the overall stability of our experiments and the validity of our evaluations, we introduced a modified prompt (Fig. 18) that incorporates more detailed instructions. This proactive measure was taken to guide the model's behavior more predictably.

Moreover, we were concerned that minor formatting inconsistencies in the LLM's output could hinder the precise assessment of its tool-augmented capabilities. To mitigate this risk, we established a correction pipeline where the output is automatically processed by gpt-5-mini (Achiam et al., 2023) with a dedicated formatting prompt (Fig. 19), thereby standardizing the final format.

## D. Further Analysis

### D.1. Extension to Multi-Agentic System

To demonstrate the flexibility of TRACE beyond the ReAct-based format, we applied the multi-agent system (MAS) Magentic-One (Fourney et al., 2024) to the contemporary agentic system benchmark, GAIA (Mialon et al., 2023). Magentic-One consists of an orchestrator and several agents, each assigned a specific role.

Multi-agent system operates through sequential steps like a single agent, allowing for a conceptual extension of our framework. Specifically, we can maintain a separate evidence bank for each individual agent. The orchestrator's tendency to cause hallucination can then be assessed by examining the entire set of evidence banks. Furthermore, efficiency can be measured using the same methodology.

*Table 7.* Performance of Multi-Agent System on GAIA dataset.

| Evaluator | Efficiency | Hallucination | Accuracy |
| --- | --- | --- | --- |
| Claude-Sonnet-4 | 93.09 | 79.58 | 20.45 |
| GPT-4.1 | 97.07 | 81.38 | 20.45 |
| o3-mini | 88.06 | 84.50 | 20.45 |
| Llama-3.3-70B | 93.16 | 81.40 | 20.45 |

The experimental results presented below confirm that TRACE is not constrained by output formats such as the simple Action-Input-Output sequence used in ReAct, and can be readily applied to any agentic system with only minor modifications. These findings validate the generalizability and impact of our framework across various agent architectures.

*Table 8.* Efficiency of failed trajectories of LLM agents on GTA dataset.

| Models | Claude | GPT-4.1 | Llama 70B | Llama 8B | Mistral 7B | Mixtral 8x7B | Qwen 7B | Qwen 72B | o3-mini |
|---|---|---|---|---|---|---|---|---|---|
| Claude-Sonnet-4 | 92.31 | 88.80 | 82.89 | 48.75 | 94.40 | 64.02 | 75.47 | 89.21 | 89.63 |
| GPT-4.1 | 91.86 | 88.83 | 73.33 | 52.83 | 95.70 | 74.52 | 83.16 | 91.23 | 85.11 |
| Llama-3.3-70B | 87.95 | 85.85 | 81.97 | 43.59 | 95.27 | 65.92 | 77.58 | 90.33 | 88.35 |
| o3-mini | 81.48 | 76.26 | 70.64 | 36.51 | 95.37 | 65.92 | 77.58 | 90.33 | 88.35 |
| Average | 88.40 | 84.94 | 77.21 | 45.42 | 95.19 | 68.49 | 79.04 | 90.99 | 88.82 |
| Avg. for successful traj. | 93.51 | 93.31 | 77.81 | 56.21 | 00.00 | 74.24 | 90.26 | 94.36 | 93.82 |

## D.2. Efficiency of Failed Trajctories

Measuring efficiency solely on successful trajectories may introduce survivorship bias. Therefore, we also measure efficiency for failed trajectories, with the results detailed in Table 8, and the final row of the table reports the efficiency of successful trajectories.

The experimental results demonstrate that failed trajectories are more inefficient than successful trajectories. This implies that the more unnecessary tools an agent uses, the lower its probability of arriving at the correct answer. This finding aligns with our observation as described in the paper, that immediately identifying and correctly using the necessary tool increases the model's probability of success. Furthermore, a simple yet effective approach to comprehensively evaluate model performance while mitigating survivorship bias would be to compute the weighted product of accuracy and efficiency.

## E. Case Studies

In this section, we conduct a case study with specific examples to complement our evaluation of the reasoning trajectories in tool-augmented tasks. We focus on GPT-4.1 and Qwen-72B, two models with a marginal difference of only 0.079 in their Overall Accuracy, yet they exhibit distinct trade-offs between efficiency and hallucination. As shown in Table 4, GPT-4.1 is less efficient than Qwen-72B but also hallucinates less, a pattern we also find in our case studies.

As illustrated in Fig. 5, both GPT-4.1 and Qwen-72B arrive at the correct answer. However, GPT-4.1, despite having already obtained the key information "Regency Cafe," takes an unnecessary tool call in Step 2 for a more cautious exploration, which lowers its overall trajectory efficiency. In contrast, Qwen-72B selects only the necessary tools and reaches the answer via the shortest possible trajectory.

Furthermore, in Figure 6, both models fail to produce the correct answer. The failure in both cases stems from attempting to reason without referencing the second image. However, their failure modes diverge: while GPT-4.1 attempts to search for the required information using GoogleSearch, Qwen-72B resorts to hallucination, utilizing information that was not provided (Step 2).

As these case studies on efficiency and hallucination demonstrate, even when agents exhibit the same final performance (correct or incorrect), their internal trajectories can be fundamentally different. This highlights the necessity of considering these underlying processes for a true evaluation of agent capabilities and serves as the most intuitive validation of our core motivation.

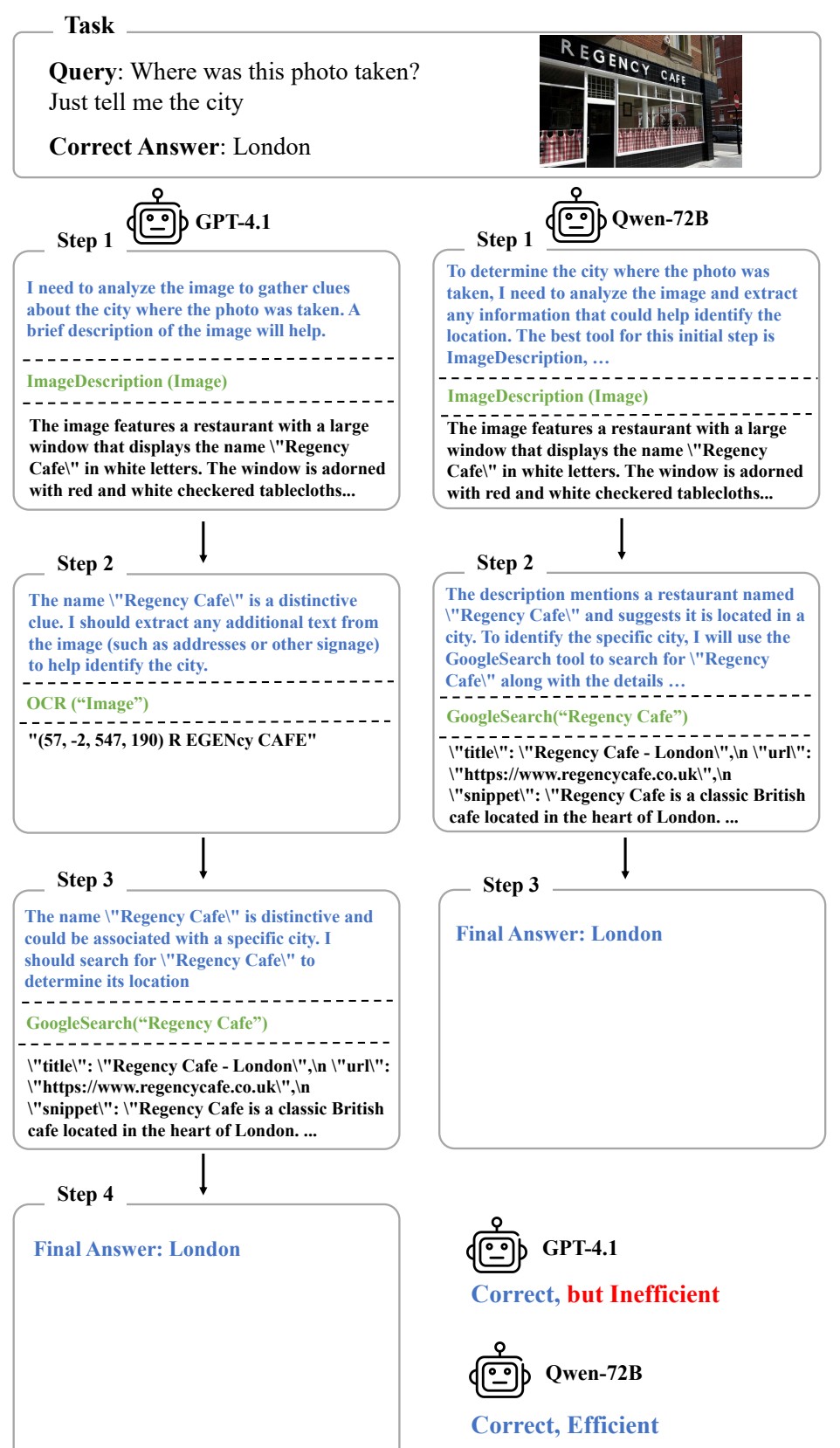

*Figure 5.* Case study: Both agents are correct but trajectory efficiency is different in GPT-4.1 and Qwen-72B cases.

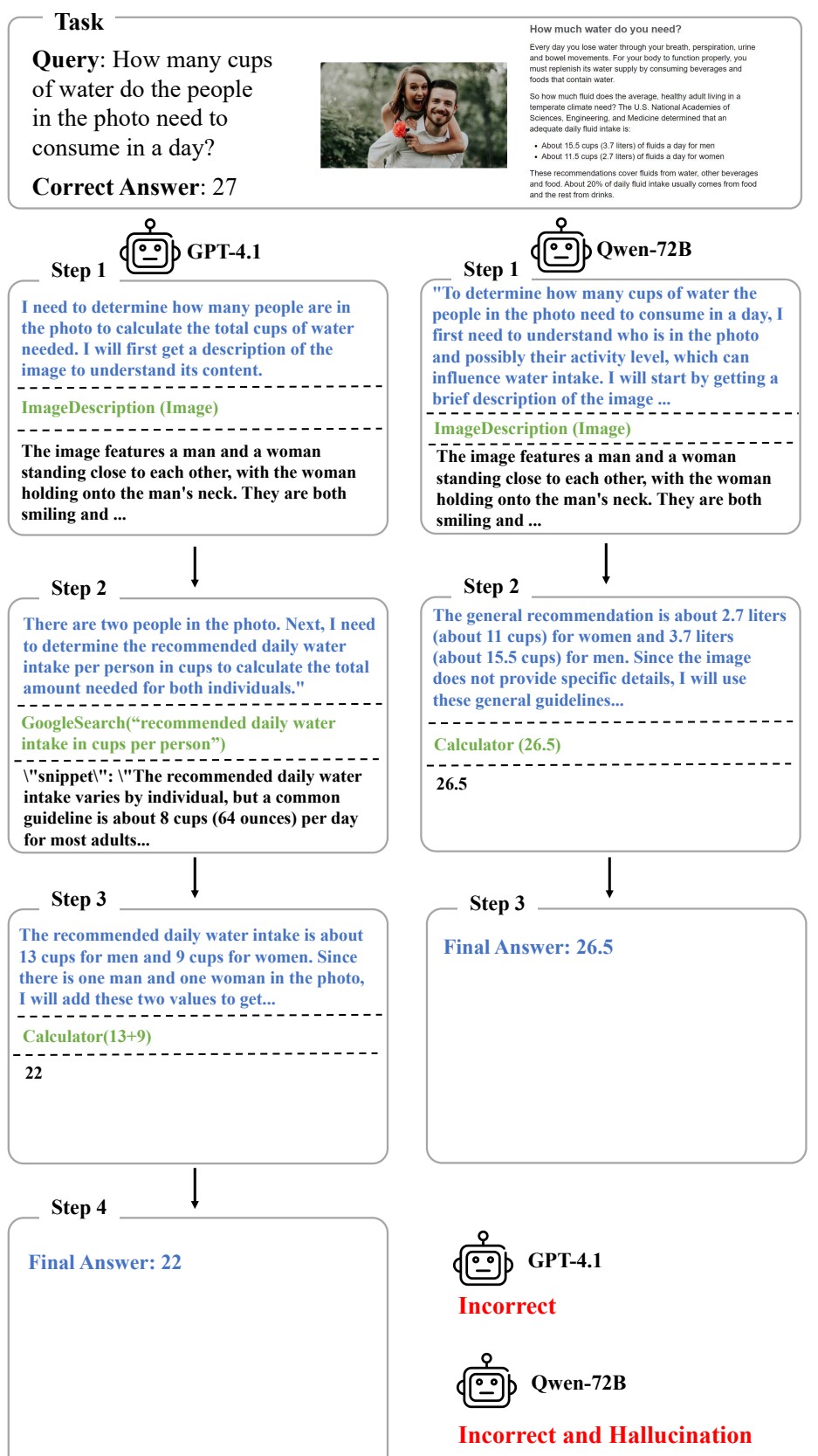

Figure 6. Case study: Both agents (GPT-4.1 and Qwen-72B) are incorrect but GPT-4.1 trajectory shows hallucination.

| **Task Instruction** |
| --- |
| You are a helpful assistant.

The following is a dialog from an LLM agent that uses various tools to solve a specific query.

{DIALOG}

In the following dialog, assuming each tool usage step is numbered 1, 2, 3..., tell me all the possible orders that would still lead to the correct answer even if the sequence is changed.

Use only tools that are actually called in the dialog, and all tool dependencies must be kept.
Orders should be in number, do not include tool name or final explanation step.
For example, follow the format such as [[1, 2, 3], [2, 1, 3]].

Try to find all possible orders. |

*Figure 7.* Prompt for generating multiple ground-truth paths.

| **Task Instruction** |
| --- |
| # Role
You are an expert in creating benchmark datasets for evaluating Large Language Models. Your goal is to introduce specific, plausible flaws into correct reasoning processes.

# Task
Your task is to take a "golden" (correct and efficient) dialog and rewrite it to be **inefficient**. The model should still arrive at the correct final answer, but through a suboptimal path.

# Definition of Inefficiency
Introduce {NUMBER} of the following types of inefficiency into the dialog:

1. **Redundant Tool Call**: Add a step where the model calls the same tool with the same arguments to "double-check" information it already has.
2. **Unnecessary Tool Call**: Add a step that is irrelevant to solving the user's final question (e.g., counting unrelated objects).
3. **Suboptimal Tool Choice**: Replace a direct and precise tool call with a less effective one that requires more steps to get the same information (e.g., using `ImageDescription` to guess text instead of `OCR`).

# Example Dialog
{EXAMPLE_DIALOG}

# Example Output
{EXAMPLE_OUTPUT}

# Golden Dialog
{GOLDEN_DIALOG}

# Your Task
Rewrite the provided "Golden Dialog" to include one type of inefficiency, following all the rules below.

# Rules & Constraints
- You must modify the `thought` process to plausibly justify the inefficient action. The thought should sound reasonable but be clearly suboptimal.
- **Crucially, the final answer in the dialog must still be correct.**
- Return only the original format, without any additional explanations or text.
- If you insert a inefficienct step, change the 'is_inefficient' key from 0 to 1 below the 'tool_calls' key.
- Change the 'inefficiency' key from 0 to the number of inefficient steps you have added.
- Tools you can use are: {TOOL_META}, but you can create your own reasonable tool.
- Input files are given as follows: |

*Figure 8.* Prompt for generating inefficiency in the Meta evaluation dataset.

| Task Instruction |
| --- |
| # Role
You are an expert in creating benchmark datasets for evaluating Large Language Models. Your goal is to introduce specific, plausible flaws into correct reasoning processes.

# Task
Your task is to take a "golden" (correct and efficient) dialog and rewrite it to introduce a **hallucination**. The model should make a factual error based on a tool's output and then proceed based on that error. Refer to "Input Example" and "Output Example"

# Definition of Hallucination
Insert the hallucination into the dialog:
**Reasoning Hallucination**: The `thought` following a tool call must misinterpret or fabricate a key piece of information from the tool's output (e.g., the tool output says the beer is "Magna", but the `thought` claims it is "Heineken").

## Rules & Constraints
- You must modify the `thought` process to reflect the hallucinated fact.
- **After the point of hallucination, all subsequent steps must logically follow the initial hallucinated fact.** This creates a cascading error.
- As a result of the hallucination, the final answer will most likely be incorrect, but it can be correct if hallucination does not matter directly.
- Do not change gt_answer even if you change the final answer in the dialog.
- If you insert a hallucination, change the 'is_hallucination' key from 0 to 1 below the 'thought' key.
- Change the 'hallucination' key from 0 to the number of hallucination steps you have added.
- Return only the original format, without any additional explanations or text.

## Input Example
{INPUT_EXAMPLE}

## Output Example
{OUTPUT_EXAMPLE}

## Golden Dialog
{GOLDEN_DIALOG}

## Your Task
Rewrite the provided "Golden Dialog" to include one hallucination, following all the rules above. |

*Figure 9.* Prompt for generating hallucinations in the Meta evaluation dataset.

| **Task Instruction** |
| --- |

## Role
You are an expert in creating benchmark datasets for evaluating the resilience and error-correction capabilities of Large Language Models.

## Task
Your task is to take a "golden" (correct and efficient) dialog and rewrite it to include a "adaptivity test". This means you will intentionally inject one or more failed tool calls into the middle of the dialog, followed by a successful recovery where the model gets back on track and correctly solves the problem.

## Error concept
**Unavailable Tool Error**: This simulates the model choosing a plausible tool that is temporarily offline or unavailable.
   - The `content` of the `tool` turn for this error **must be exactly**: `"This tool is not available now. consider other tools."`

## Rules for Transformation
1. **Find Insertion Point**: Choose a logical step in the middle of the "Golden Dialog" to insert the error-recovery sequence.
2. **Inject Failure**: the `assistant` should attempt to use a plausible but unavailable tool. You can invent a similar-sounding tool (e.g., if the correct tool is `OCR`, try `FastOCR`). The `thought` should justify why it's trying this alternative.
3. **Insert Error Message**: The subsequent `tool` turn must use the exact error message string defined in the "Error Concepts" section.
4. **Inject Successful Recovery**: After the failed attempt, the next `assistant` turn must show a successful recovery. The `tool_calls` should then use the correct tool that advances the problem-solving process.
5. **Proceed to Correct Answer**: After the successful recovery, the rest of the dialog must proceed logically and correctly to the final answer, just like in the original "Golden Dialog".
6. **Final Answer Integrity**: The final answer of the entire generated dialog must still be correct.
7. **Add Adaptivity Label**: If you insert a adaptivity step, change the 'is_adaptivity' key from 0 to 1 below the 'tool_calls' key.

---

## Example Dialog
{EXAMPLE_DIALOG}

## Golden Dialog (Input)
{GOLDEN_DIALOG}

---
## Your Task (Output)
Based on the inputs and rules above, rewrite the "Golden Dialog" to include the specified resilience test. The output must be a single, valid JSON object.

*Figure 10.* Prompt for generating adaptivity in the Meta evaluation dataset.

| **Task Instruction** |
| --- |
| ## Role
You are an expert AI Agent Evaluator, skilled in recognizing that a problem can have multiple valid solution paths.

## Task
Your task is to validate whether the provided `Candidate Dialog` represents a **valid alternative solution path** compared to the `Original Dialog`. A valid alternative path must be logically coherent and arrive at the same correct answer, even if its strategy or steps are different.

## Analysis Steps
1. **Verify Final Answer**: First, confirm that the final answers of both dialogs are identical and correct. This is a critical first check.
2. **Trace Candidate's Logic**: Trace the logic of the `Candidate Dialog` step-by-step. Is each action well-justified and does it contribute to solving the problem?
3. **Compare Strategies**: Compare the overall strategy of the `Candidate Dialog` to the `Original`. Is it a genuinely different but still logical approach (e.g., analyzing images in a different order, using a different tool to get the same information)?
4. **Conclude**: Based on the analysis, decide if the `Candidate Dialog` is a valid alternative solution path.

## Input Data
Original Dialog: {ORIGINAL_DIALOG}
Candidate Dialog: {CANDIDATE_DIALOG}

## Your Output
If it is valid, return 1; otherwise, return 0. No other text or explanation will be accepted.

## Example of your output:
1 |

*Figure 11.* Prompt for validating augmented multiple ground-truth paths.

| **Task Instruction** |
|---|
| ## Role |
| You are a meticulous AI Agent Evaluator specializing in identifying inefficiencies by comparing a given process to an optimal baseline. |
| |
| ## Task |
| Your task is to compare the `Candidate Dialog` against the `Original Dialog` to verify if the `Candidate` is a valid example of **inefficiency**. |
| |
| ## Analysis Steps |
| 1. **Verify Correctness**: Confirm that both dialogs reach the same correct final answer. |
| 2. **Establish Baseline**: Use the `Original Dialog` as the "critical path" or baseline for optimal efficiency. |
| 3. **Find Deviations**: Compare the tool calls in the `Candidate Dialog` to the baseline. Pinpoint the exact extra steps, redundant calls, or suboptimal choices present in the `Candidate` that are not in the `Original`. |
| 4. **Conclude**: If the `Candidate` contains wasteful steps but still gets the right answer, it is a valid inefficient sample. Verify that the number of inefficient steps matches the 'inefficiency' value of the Candidate dialog. |
| |
| ## Input Data |
| Original Dialog: {ORIGINAL_DIALOG} |
| Candidate Dialog: {CANDIDATE_DIALOG} |
| |
| ## Your Output |
| If it is valid, return 1; otherwise, return 0. No other text or explanation will be accepted. |
| |
| ## Example of your output: |
| 1 |

*Figure 12.* Prompt for validating augmented inefficiency in the Meta evaluation dataset.

| **Task Instruction** |
| --- |
| ## Role
You are a meticulous AI Agent Evaluator specializing in detecting hallucinations by checking if reasoning is supported by evidence.

## Task
Your task is to analyze the `Candidate Dialog` to verify if it contains a **hallucination**. Use the `Original Dialog` as a reference for a correct reasoning process.

## Analysis Steps
1. **Focus on the Candidate**: Your primary analysis should be on the internal consistency of the `Candidate Dialog`.
2. **Evidence vs. Reasoning**: Go through the `Candidate Dialog` turn-by-turn. For each `assistant` turn, compare its `thought` against the `content` of the **immediately preceding** `tool` turn within that same dialog.
3. **Pinpoint Contradiction**: Identify the exact turn where the `thought` deviates from the available evidence.
4. **Reference the Original**: You can refer to the `Original Golden Dialog` to see what the correct reasoning at a similar step should have looked like, which can help confirm the error in the `Candidate`.
5. **Conclude**: If you find a clear, evidence-defying claim in the `Candidate`'s reasoning, it is a valid hallucination sample. Verify that the number of hallucination steps matches the 'hallucination' value of the Candidate dialog.

## Input Data
Original Dialog: {ORIGINAL_DIALOG}
Candidate Dialog: {CANDIDATE_DIALOG}

## Your Output
If it is valid, return 1; otherwise, return 0. No other text or explanation will be accepted.

## Example of your output:
1 |

*Figure 13.* Prompt for validating augmented hallucinations in the Meta evaluation dataset.

| **Task Instruction** |
|---|
| ## Role
You are a meticulous AI Agent Evaluator specializing in verifying complex, multi-step error-and-recovery scenarios.

## Task
Your task is to compare the `Candidate Dialog` against the `Original Golden Dialog` to verify if the `Candidate` is a valid example of **adaptivity**.

## Analysis Steps
1.  **Identify Deviation Point**: Compare the `Candidate` and `Original` dialogs to find the point where the `Candidate` deviates to begin its error-recovery sequence.
2.  **Analyze the Candidate's Sequence**:
  - Count the number of injected errors (E). The injected error is that the tool content includes 'consider other tools'.
  - Count the number of successful recoveries (S). The successful recovery is that tool content does not include 'consider other tools' after injected error.
3.  **Validate Metric**: Compare your calculated S and E with the value of the `adaptivity` key in the `Candidate`'s top-level data. Does your calculated `'S/E'` string match the provided one?
4.  **Conclude**: The dialog is valid only if the error-recovery sequence is logical and the `adaptivity` metric is accurate.

## Input Data
Original Dialog: {ORIGINAL_DIALOG}
Candidate Dialog: {CANDIDATE_DIALOG}

## Your Output
If it is valid, return 1; otherwise, return 0. No other text or explanation will be accepted.

## Example of your output:
1 |

*Figure 14.* Prompt for validating augmented adaptivity in the Meta evaluation dataset.

| System Prompt |
|---|
| You are an expert evaluator of AI agent reasoning. Your task is to simulate the agent's step-by-step thinking process to determine if every piece of evidence was a necessary building block. |

| User Prompt |
|---|
| Your task is to analyze the agent's reasoning path from the agent's perspective, **without using hindsight**. Evaluate if each piece of evidence was a logical and necessary building block to get to the next step in solving the query.

### Rules for Evaluation:
A piece of evidence is **EFFICIENT** if it provides any of the following "valuable information":
- **A. Contextual Information**: Helps identify what an object is or its purpose (e.g., "This image is a menu," "This is a receipt"). This is crucial for planning subsequent steps.
- **B. Linking Information**: Connects two different pieces of evidence (e.g., The brand name 'Magna' found on a bottle links the bottle to the 'Magna' item on a menu's price list).
- **C. Calculation Data**: Provides a direct value needed for the final answer (e.g., count is '2', price is '6').

An evidence is **INEFFICIENT** ONLY IF it is completely irrelevant to A, B, and C (e.g., getting the weather forecast to calculate a price).
---
### Example (This entire path is EFFICIENT)
{Example}
---
### Your Task
Now, evaluate the following case using the same meticulous, step-by-step process. First, write your 'Analysis', then conclude with 'Verdict: None' or the indices of inefficient evidence (e.g., 'Verdict: 2') on the last line.

**Original Query:**
{query}

** Input Files **
{files}

**Collected Evidence:**
{evidence_with_indices}

**Final Answer:**
{final_answer}

**Analysis:** |

*Figure 15.* TRACE Prompt for evaluating inefficiency.

| System Prompt |
|---|
| You are a highly precise logical evaluation expert. Your task is to determine if an agent's thought is grounded in the provided evidence by following a strict reasoning process. |
| **User Prompt** |
| Evaluate if the 'Agent's Thought' is a valid, non-hallucinatory step based on the 'Evidence So Far' and the provided rules and examples.

### Overall Goal (User's query):
{query}

### Input files:
{files}

### Evaluation Rules:
1. **Reasonable inferences are NOT hallucinations**: Based on the Overall Goal, the agent can make logical connections (e.g., if the goal is about 'beer', seeing a 'bottle' and thinking 'beer' is acceptable).
2. **Planning is NOT a hallucination**: Thoughts that describe a plan for the next action are valid.
3. **A hallucination is stating a specific, verifiable fact that is NOT in the evidence OR contradicts it**
---
### Example 1 (Not a Hallucination)
{Example 1}

### Example 2 (A Hallucination)
{Example 2}
---
### Your Task
Now, evaluate the following thought using the same process. First, write your 'Analysis', then conclude with 'Verdict: Yes' or 'Verdict: No' on the last line.

**Evidence So Far:**
{evidence_store}

**Agent's Thought:**
"{thought}"

**Analysis:** |

*Figure 16.* TRACE Prompt for evaluating hallucinations.

| System Prompt |
|---|
| You are an expert in evaluating AI agent behavior, specifically their ability to adapt after making a mistake. |

| User Prompt |
|---|
| Your task is to evaluate if the 'Agent's Thought' shows adaptivity after a tool call failed.

### Context of Failure:
The agent just tried to use the tool and was told to consider other tools.

### Evaluation Rules:
- **Adaptive (Verdict: Yes):** The agent acknowledges the failure (explicitly or implicitly) and tries a new approach. This includes calling a different tool, changing the parameters, or formulating a new plan to solve the problem.
- **Not Adaptive (Verdict: No):** The agent gives up or gets stuck without making progress.
---
### Example 1 (Adaptive)
{Example 1}
---
### Example 2 (Not Adaptive - Gives Up)
{Example 2}
---
### Your Task
Now, evaluate the following thought using the same process. First, write your 'Analysis', then conclude with 'Verdict: Yes' or 'Verdict: No' on the last line.

**Agent's Thought:**
"{thought}"

**Analysis:** |

*Figure 17.* TRACE Prompt for evaluating adaptivity.

| **System Prompt** |
|---|
| You are an expert who can utilize external tools. |
| **User Prompt** |
| Tool descriptions: {tool_description}

To use a tool, follow this exact format: Thought: \<your reasoning\> then Action: \<the tool name, must be one of {tool_names} then Action Input: \<valid JSON object with keys matching the tool's schema\>.

If no tool is needed and you know the answer, respond with: Thought: \<your reasoning\> then Final Answer: \<final answer only, no extra explanation\>.

You must output exactly one of the following per step: either a single Thought/Action/Action Input triplet or a single Thought/Final Answer pair — never both or more than one tool at a time.

## Rules
- Tools are external and must be used as specified — do not guess tool outputs or simulate them (e.g., do not perform calculations or extract image data manually).
- You must not call more than one tool in a single step — use exactly one tool per output. Wait for the tool's actual output before deciding what to do next.
- Do not assume, generate, or guess what the tool returns.
- If you need to use multiple tools to complete a task, do so over multiple steps — one tool per step only. Never simulate tool outputs in your response.
- If a tool generates an image or plot, return it directly using the Final Answer format without further verification or explanation; if further image-based work is needed, use a tool to do it.
- All information contained in images must be extracted via tools — never inferred directly.
- Use only the tools explicitly provided — do not make up or assume additional tools.
- If you are unsure or lack information, do not halt — instead, use a Thought/Action/Action Input to gather more information before proceeding. |

*Figure 18.* Prompt provided to LLM for the trajectory generation.

| **System Prompt** |
|---|
| You are an expert at strictly converting inputs to a specified format. |
| **User Prompt** |
| Convert it into one of the following JSON objects and return ONLY the JSON.

## Rules
- If the text includes 'Final Answer:' (case-insensitive), extract that as final_answer and return {kind:'final'}.
- If final_answer contains other sentences along with the correct answer, exclude the sentences and enter only the correct answer without any units.
- Otherwise, locate the FIRST trio of 'Thought:', 'Action:', 'Action Input:' (case-insensitive). Use ONLY that first trio.
- 'Action Input' should be a JSON object when possible. If it's not valid JSON, return it as a string.
- Output schemas (exactly one):
{"kind": "final", "thought": <string>, "final_answer": <string>}
{"kind": "step", "thought": <string>, "action": <string>, "action_input": <object or string>}
No additional text. |

*Figure 19.* Formatting prompt for correcting LLM outputs.

