# OpenReview forum: "Beyond the Final Answer: Evaluating the Reasoning Trajectories of Tool-Augmented Agents"
_ICML.cc/2026/Conference — ICML 2026 regular_

### Official Review · Reviewer_sLwe · 2026-03-10

**Soundness:** 3
**Presentation:** 2
**Significance:** 3
**Originality:** 3
**Overall Recommendation:** 5
**Confidence:** 4

**Summary:**

This paper introduces TRACE, a framework for evaluating tool-augmented LLMs across three dimensions -- inefficiency, hallucination, and adaptivity -- without requiring human-annotated ground truth. Efficiency measures unnecessary evidence collection, hallucination detects when an agent's reasoning deviates from gathered evidence, and adaptivity assesses how the agent handles unavailable tools; all three are evaluated by an LLM rather than against pre-defined trajectories. Central to the framework is an "evidence bank," a knowledge base of facts the agent collects during a task. The authors also introduce two meta-evaluation datasets, Meta-GTA and Meta-M&M, and demonstrate that TRACE accurately evaluates agent trajectories on both. The authors release an open-source implementation of TRACE.

**Compliance With Llm Reviewing Policy:**

Affirmed.

**Key Questions For Authors:**

1. Could the authors provide a token-level measure of evaluator cost (e.g., input/output tokens consumed per evaluation) to allow for a more reproducible and hardware-agnostic comparison across evaluator models?

2. To what extent might LLM evaluators systematically favor reasoning styles similar to their own training data, and have the authors considered whether evaluator-agent model overlap (i.e. using Claude to evaluate Claude) introduces any measurable bias in the reported results?

3. The authors apply TRACE to real agent trajectories on the GTA dataset in Section 5. I'm curious to what extent does the type and distribution of errors introduced in your meta-evaluation dataset reflect those observed in trajectories produced by real deployed systems. Have the authors validated that TRACE's strong performance on Meta-GTA and Meta-m&m's transfers to in-the-wild uncontrolled failure modes?

**Limitations:**

No, see limitations for feedback on the impact statement.

**Strengths And Weaknesses:**

**Strengths**
- **Reference-free evaluation.** Reference-free evaluation removes the need for human-annotated ground truth, which is a significant practical advantage given how expensive and restrictive single ground-truth trajectories are in practice.
- **Robust.** Robustness to multiple valid paths is a clever design choice; the evidence bank evaluates whether steps were necessary rather than whether they match a pre-defined sequence, which makes the framework more realistic and flexible.
- **Cost-effective.** The authors show that, with their method, Llama-3.3-70B performs comparably to proprietary models.
- **Extensible.** The approach is extensible to diverse agentic architectures, including single ReAct agents and multi-agent systems.
- **Well-written.** The paper is generally well-written and easy to follow.

**Weaknesses**
- **Efficiency of LLM evaluators is not well-evaluated.** Wall-clock time is an insufficiently rigorous measure of evaluator efficiency in Figure 3, as it is sensitive to exogenous factors such as API latency and infrastructure load; a token-level measure of evaluator cost would be more reproducible and hardware-agnostic.
- **Presentation of results.** The results could be presented better. Table 1 has redundant information, and displaying both raw LLM-as-a-Judge scores and TRACE deltas in parentheses makes the table harder to read than either representation would be on its own. Tables 2 and 3 address closely related claims and would be more effectively communicated as a single consolidated figure that presents both accuracy and variance together. Table 4 is difficult to parse in its current form (results for four evaluators across nine models are laid out in a flat grid).
- **Weak impact statement.** The impact statement does not meaningfully engage with the societal implications of automated agent evaluation. A more substantive statement might briefly acknowledge that frameworks like TRACE could accelerate the deployment of tool-augmented agents in high-stakes settings, and that LLM-based evaluators may inherit biases that systematically favor certain reasoning styles or penalize valid but unconventional trajectories.

---

> ### Author Rebuttal · Authors · 2026-03-30
>
> We would like to express our sincere gratitude to the reviewer for their insightful evaluation and valuable suggestions. We especially appreciate the reviewer’s acknowledgment of the diverse strengths of our work. In the following, we provide a point-by-point response to the specific concerns and questions raised by the reviewer.
>
> ---
>
> ### **W1. & Q1.** ###
>
> We agree that wall-clock time is sensitive to exogenous factors, and we provide the token-level evaluation cost as a more reproducible and hardware-agnostic measure.
>
> The table below reports the average output tokens consumed per evaluation query for each evaluator model. Note that input tokens are identical across all evaluators, as TRACE uses the same standardized prompt structure for all models.
>
> | |min|max|average|
> |-|-|-|-|
> |Claude|728|2,756|1,305.7|
> |GPT-4.1|348|1,479|733.34|
> |o3-mini|1,402|6,341|3,164.8|
> |Llama 70B|547|2,158|1,041.16|
> |Llama 8B|439|1,803|991.3|
>
> GPT-4.1 is the most token-efficient evaluator, while o3-mini consumes substantially more tokens due to its nature as a reasoning model, which generates extended internal chain-of-thought reasoning. The remaining models fall within a comparable range.
>
> Importantly, token consumption alone does not determine the practical value of an evaluator. For instance, Llama-3.3-70B achieves evaluation accuracy comparable to larger proprietary models (Table 1) while consuming only 1,041 output tokens per query, making it a highly cost-effective choice. Conversely, while o3-mini consumes the most tokens, it also delivers strong evaluation performance, reflecting the additional computational effort of its reasoning process.
>
> ---
>
> ### **Q2.** ###
>
> We thank the reviewer for this important concern. To investigate whether evaluator-agent model overlap introduces bias, we analyzed the results in Table 4 where the same model family serves as both agent and evaluator. We observe no evidence of systematic self-preferencing, as evaluators do not consistently assign higher scores to agents from their own model family compared to other evaluators.
>
> Furthermore, we present the Fleiss' kappa scores across evaluators below, indicating substantial inter-evaluator agreement which means that evaluation outcomes are largely consistent regardless of which model is used as the evaluator. These findings provide evidence that TRACE's structured, evidence-grounded evaluation mitigates potential biases arising from evaluator-agent overlap.
>
> |  | Efficiency | Hallucination | Adaptivity |
> |-|-|-|-|
> | Fleiss' Kappa | 0.4928| 0.6112| 0.6723|
>
> ---
>
> ### **Q3.** ###
>
> The error types in our meta-evaluation dataset were not designed arbitrarily but were informed by patterns observed in real agent trajectories. Specifically, we first examined trajectories produced by various LLM agents on the GTA and m&m's benchmarks and identified recurring failure patterns such as redundant or unnecessary tool calls (inefficiency), reasoning that deviates from tool outputs (hallucination), and failure to recover from unavailable tools (lack of adaptivity). We then systematically reproduced these patterns through controlled injection (detailed in Appendix B.1), ensuring that the meta-evaluation errors reflect realistic failure modes rather than artificial edge cases.
>
> Furthermore, the case studies in Appendix E illustrate that TRACE identifies diverse in-the-wild failure modes including unnecessary tool calls by GPT-4.1 and hallucinations by Qwen-72B, that were not explicitly modeled in the meta-evaluation dataset. This provides qualitative evidence that TRACE generalizes beyond the specific error patterns it was validated on.
>
> To further substantiate that TRACE's performance transfers to uncontrolled settings, we conducted a human evaluation study on the full set of real agent trajectories used in Section 5. Human annotators independently assessed the efficiency, hallucination, and adaptivity of all trajectories generated by actual LLM agents, and we measured the agreement between human judgments and TRACE's evaluations. The results as shown below confirm that TRACE's assessments align closely with human evaluations on in-the-wild trajectories, validating its applicability beyond the controlled meta-evaluation setting.
>
> | |Human agreement|
> |-|-|
> |Efficiency|98.76|
> |Hallucination|95.93|
> |Adaptivity|99.98|
>
> ---
>
> ### **W2.** ###
>
> We thank the reviewer for the constructive feedback on the presentation. In the revised manuscript, we will reorganize the tables as suggested: simplifying Table 1 to avoid redundant information, consolidating Tables 2 and 3 into a single figure, and restructuring Table 4 for easier parsing.
>
> ---
>
> ### **W3.** ###
>
> We thank the reviewer for this suggestion. We will revise the impact statement including the potential for TRACE to accelerate deployment of tool-augmented agents in high-stakes settings, as well as the risk that LLM-based evaluators may inherit biases that systematically favor certain reasoning styles.

---

> > ### Author Rebuttal · Reviewer_sLwe · 2026-04-02
> >
> > Thank you for your response; I retain my positive assessment of the paper. I expect that the results about wall-clock time (Q1), evaluator-agent overlap (Q2), and the human evaluation study (Q3) will be included in the main paper.

---

### Official Review · Reviewer_hQMh · 2026-03-11

**Soundness:** 2
**Presentation:** 2
**Significance:** 2
**Originality:** 2
**Overall Recommendation:** 4
**Confidence:** 3

**Summary:**

This paper introduces the reference-free framework TRACE, a tool to enhance the multidimensional evaluation of LLMs. By merging an evidence base that accumulates knowledge from the previous steps, TRACE can effectively evaluate the agent's reasoning trajectory. To verify the framework, the authors developed a new meta-evaluation dataset with diverse and defective tracks, each marked with multiple performance scores.

**Compliance With Llm Reviewing Policy:**

Affirmed.

**Key Questions For Authors:**

1. Trade-off between the evaluation metrics.
There may be a trade-off between the evaluation metrics proposed by the authors, which needs more comprehensive and in-depth discussion. For example, if a tool fails and two or more other tools are required to cooperate in the alternative scheme to complete the work of the failed tool, will the scheme replacement for high adaptivity affect the calculation of the efficiency score?

2. Dependence on evaluator intelligence.
In this paper, the evaluator is used to determine the minimum evidence subset, but if the evaluator cannot identify the dependencies between some steps, it mistakenly regards some necessary steps as redundant steps, which will affect the calculation of efficiency. The author should analyze the impact of the evaluator more deeply.

**Limitations:**

1. Limitations of the definition of hallucination.
The article believes that if the result of thinking deviates from the facts in the evidence bank, then there is a hallucination. But models often need to combine internal knowledge outside the evidence base, such as common sense. The definition of hallucination in this paper may be too strict.


2. Adaptability test scenarios lacking diversity.
In this paper, the adaptability test is carried out by replacing the available tool name with the unavailable tool name, but in reality, there are many scenarios of tool failure, such as the tool returning empty or wrong results, and the current test methods can not fully reflect the various situations of tool failure.

3. Other Issues
1) Spelling mistakes.
"Emtpy" in the left column of line 201 may be misspelled. The author may want to use "Statistics" in line 630.

2) Redundant words.
Two consecutive "metric" appear in the right column of line 190.

**Strengths And Weaknesses:**

Strengths:

1. This paper proposes an effective evaluation framework focusing on the quality of reasoning trajectory, introduces three quantifiable evaluation dimensions, and provides a supporting meta-evaluation dataset.

2. This paper analyzes the performance of the LLM agent in complex tool tasks and puts forward specific strategies to improve its reasoning and reliability.


Weaknesses:

1. Limitations of the definition of hallucination.

2. Adaptability test scenarios lacking diversity.

3. More comprehensive and in-depth analysis required.

4. Other issues.

---

> ### Author Rebuttal · Authors · 2026-03-30
>
> We appreciate the reviewer’s constructive critique, which has helped improve the quality of our manuscript. Specifically, we are grateful that the reviewer recognized the effectiveness of our framework. We have carefully addressed each of the limitations and questions identified by the reviewer.
>
> ---
>
> ### **W1. & L1.** ###
>
> A core motivation of tool-augmented agents is to overcome the limitations of **LLMs' internal knowledge which may be outdated or incorrect**, by leveraging external tools to obtain up-to-date and reliable information. In this paradigm, the agent is expected to ground its reasoning on the evidence gathered through tool interactions rather than relying on internal knowledge. If an agent bypasses tool outputs and instead draws on internal knowledge, it introduces the risk of using information that is factually incorrect or outdated. For instance, an agent might use internally stored knowledge about a restaurant's location, which may have changed, leading to incorrect conclusions despite seemingly reasonable reasoning.
>
> For this reason, we adopt a strict definition of hallucination that flags any thought not grounded in the evidence bank, aligning with recent work in the tool-augmented agent literature [1].
>
> [1] Li, Shiyu, et al. "ReSeek: A Self-Correcting Framework for Search Agents with Instructive Rewards." arXiv preprint arXiv:2510.00568 (2025).
>
> ---
>
> ### **W2. & L2.** ###
>
> To demonstrate the generalizability of our adaptivity metric, we extended our evaluation to include **513 new samples covering a wider range of failure scenarios**. Specifically, we tested the model's ability to recover from:
>
> - Timeout errors (e.g., 'Error 408: Tool running time out')
> - Rate-limiting errors (e.g., 'Error 429: Too many requests')
> - Empty outputs (where the tool executes successfully but returns no data)
>
> Results presented below show that TRACE maintains high evaluative performance across these diverse failure types. This confirms that the metric is scalable and applicable whenever a tool fails to operate correctly, regardless of the specific error encountered.
>
> | |New adaptivity|
> |-|-|
> |Claude|98.62|
> |GPT-4.1|96.83|
> |o3-mini|97.22|
> |Llama 70B|97.6|
> |Llama 8B|87.05|
>
> Regarding the scenario of tools returning incorrect results, we consider this to fall outside the scope of our current adaptivity definition, as detecting incorrect outputs requires a fundamentally different capability which is distinct from the ability to recover from explicit tool failures. We believe this represents an interesting and complementary direction for future work.
>
> ---
>
> ### **W3. & Q1.** ###
>
> If a tool fails and the agent adaptively selects two or more alternative tools to accomplish the same goal, the failed tool call is counted as unnecessary evidence, while the successful recovery is credited as adaptive behavior. In this case, the agent would receive a high adaptivity score but a slightly lower efficiency score compared to an agent that directly selected the correct tool without encountering any failure.
>
> We argue that this behavior is not a flaw in our metrics but rather an accurate reflection of the agent's actual performance. An agent that encounters a tool failure and recovers is, in reality, less efficient than one that succeeds on the first attempt even though the former demonstrates greater resilience. Capturing this nuance is precisely why multi-dimensional evaluation is necessary since a single metric would obscure these meaningful differences.
>
> Indeed, this trade-off pattern is observable in our experimental results. For instance, in Table 4, certain models achieve high adaptivity scores while showing relatively lower efficiency, reflecting scenarios where successful recovery comes at the cost of additional steps.
>
> ---
>
> ### **W3. & Q2.** ###
>
> If the evaluator were misidentifying necessary steps as redundant due to failure in capturing inter-step dependencies, this would directly manifest as low efficiency accuracy in our meta-evaluation. Our meta-evaluation dataset contains explicitly labeled inefficient steps, and accuracy is measured against these ground-truth annotations. As shown in Table 1, the efficiency accuracy achieved by TRACE is consistently high across evaluator models, indicating that evaluators are successfully identifying the dependencies between steps and distinguishing necessary evidence from redundant evidence.
>
> We also acknowledge that evaluator capability does affect performance, as expected. As shown in Table 1, stronger evaluator models achieve higher efficiency accuracy, suggesting that more capable models are better at recognizing subtle dependencies between steps. However, even smaller models (e.g., Llama-3.3-70B) achieve competitive performance under TRACE, substantially outperforming the naive LLM-as-a-Judge approach.
>
> ---
>
> ### **W4. & L3.** ###
>
> We would like to thank the reviewer for their thorough review. We have revised the manuscript accordingly to reflect these suggestions.

---

> > ### Author Rebuttal · Reviewer_hQMh · 2026-04-02
> >
> > Thank you for your detailed reply. The author made a clear explanation of my questions about hallucination and matrix logic, and added sufficient experiments to solve my doubts about adaptive scenarios. I decide to maintain the original positive evaluation, and expect the author to implement the following modifications involved in the reply in the final version:
> > 1. Add a discussion on the trade-off between efficiency and adaptability to the text.
> > 2. Supplement the rationality of the definition of hallucination in the text;
> > 3. Add the supplementary experiment in the reply to the text or the appendix.

---

### Official Review · Reviewer_H4m4 · 2026-03-13

**Soundness:** 3
**Presentation:** 3
**Significance:** 2
**Originality:** 3
**Overall Recommendation:** 4
**Confidence:** 4

**Summary:**

The paper introduces TRACE, a reference-free evaluation framework that assesses tool-augmented agent reasoning trajectories via an evidence bank along the dimensions of efficiency, hallucination, and adaptivity.

**Compliance With Llm Reviewing Policy:**

Affirmed.

**Final Justification:**

The rebuttal addresses my concerns. I am increasing the overall assessment score.

**Key Questions For Authors:**

Please refer to the weakness section.

**Limitations:**

Overall, TRACE provides a compelling trajectory-level evaluation framework, but its reliance on LLM judges, subjective efficiency criteria, and limited reliability analysis constrains its generalizability.

**Strengths And Weaknesses:**

### Summary of strengths:
1. While prior work has explored hallucination detection and tool-use efficiency in isolation, the paper’s unified, multi-dimensional evaluation of efficiency, hallucination, and adaptivity for agentic trajectories is novel.
2. The paper is well written and easy to follow.

### Summary of weaknesses:
1. The paper relies heavily on LLM‑based judges for scoring. While this approach helps reduce the cost of annotating multiple possible trajectories per request, it does not adequately account for the inherent biases, calibration issues, and task‑specific limitations of the LLM‑as‑a‑judge itself.
2. While the paper compares TRACE against trajectory‑based benchmarks and a vanilla LLM‑as‑a‑judge using accuracy across three dimensions, this evaluation is incomplete. In the absence of an objective ground truth, accuracy can be misleading. Reporting an agreement metric such as Cohen’s kappa would be more appropriate to quantify consistency beyond chance and to substantiate the claimed improvements.
3. I do not fully understand why LLM‑as‑a‑judge underperforms, nor why TRACE benefits from the introduction of an evidence bank. The evidence bank seems to consist of the same (action, input, output) information that is already available in the agent’s trajectory presented to the judge model. If both representations contain the same underlying information, what structural or semantic difference in the evidence bank enables TRACE to leverage it more effectively?

---

> ### Author Rebuttal · Authors · 2026-03-30
>
> We are deeply grateful to the reviewer for their thorough review and valuable feedback. We particularly thankful that the reviewer admitted the novelty of unified multi-dimensional evaluation. We're happy to address each of the weaknesses that the reviewer raised.
>
> ---
>
> ### **W1.** ###
>
> We acknowledge that reliance on LLM-based judges is a shared challenge across the evaluation community, and we address the three specific issues raised.
>
> **Regarding bias.** To examine systematic bias, we analyzed the evaluation results in Table 4 where the same model family serves as both the agent and the evaluator (e.g., Claude evaluating Claude). We observe **no evidence of self-preferencing bias**, i.e., evaluators do not consistently assign higher scores to agents from their own model family. This suggests that TRACE's structured evaluation protocol that is grounded in the evidence bank helps anchor the LLM's judgement on factual evidence rather than stylistic familiarity.
>
> **Regarding calibration.** A key design principle of TRACE is to improve the calibration and consistency of LLM-based evaluation by replacing unstructured dialog-level assessment with structured, evidence-grounded judgment. Rather than asking an LLM to assess an entire trajectory holistically, TRACE provides the evaluator with a curated evidence bank that incrementally accumulates factual information from each step. This structured input reduces the ambiguity in the evaluator's task and leads to more consistent judgments. As shown in Table 3, TRACE achieves **lower standard deviation** across multiple valid trajectories compared to PIPA, indicating more stable and well-calibrated evaluation.
>
> **Regarding task-specific limitations.** TRACE is designed to be task-agnostic: the evidence bank mechanism operates over any tool-interaction sequence regardless of the specific task domain. To validate this generalizability, we applied TRACE not only to standard ReAct-style agents but also to a **multi-agent system (Magentic-One) on the GAIA benchmark** (Appendix D.1), confirming its applicability across diverse agent architectures and task types.
>
> Finally, to validate the reliability of TRACE as an evaluator, we conducted a study comparing TRACE's assessments with human judgments on the full set of real trajectories (Agent model: Claude Sonnet, Evaluator model: GPT-4.1) as shown below. These results, spanning all collected samples, demonstrate that although TRACE is LLM-based, it can evaluate as effectively as humans across diverse real-world scenarios.
>
> | |Human agreement|
> |-|-|
> |Efficiency|98.76|
> |Hallucination|95.93|
> |Adaptivity|99.98|
>
> ---
>
> ### **W2.** ###
>
> Following the reviewer's suggestion, we computed agreement metrics across all evaluator models. Since our evaluation involves more than two evaluators, we adopt Fleiss' kappa, a generalization of Cohen's kappa to the multi-rater setting. The results are as below:
>
> | |Efficiency|Hallucination|Adaptivity|
> |-|-|-|-|
> |Fleiss' Kappa|0.4928|0.6112|0.6723|
>
> All three metrics demonstrate meaningful agreement well above chance [1], indicating that TRACE produces consistent evaluations regardless of which model is used as the evaluator.
>
> We also note that, while the reviewer rightly points out the challenge of evaluation without objective ground truth, our meta-evaluation dataset does in fact contain ground-truth labels. Each step is explicitly annotated as efficient/inefficient, hallucinatory, or adaptive based on controlled injection. Thus, the accuracy reported in Table 1 is measured against known labels, making it a meaningful metric in this controlled setting.
>
> [1] Landis, J. Richard, and Gary G. Koch. "The measurement of observer agreement for categorical data." biometrics (1977): 159-174.
>
> ---
>
> ### **W3.** ###
>
> While the evidence bank and the raw trajectory contain the same underlying information, the key difference lies in **how** this information is structured and presented to the evaluator LLM.
>
> In the naive LLM-as-a-Judge approach, the entire unstructured dialog is provided to the evaluator, which must then independently identify relevant information, track factual consistency, and assess efficiency across all steps simultaneously.
>
> In contrast, the evidence bank decomposes the trajectory into structured (action, input, output) tuples, providing the evaluator with clearly organized factual records rather than raw conversation. This reduces the cognitive burden on the model by eliminating the need to parse and extract relevant information from unstructured dialog.
>
> This structural advantage is particularly pronounced for longer trajectories. While the average token count per trajectory in the Meta-GTA dataset is 806, the average number of tokens for the trajectories that LLM-as-a-Judge failed was 863. In contrast, the average token count for the trajectories that TRACE failed was 811. This suggests that TRACE can effectively evaluate even longer trajectories, leveraging the structured evidence bank.

---

> > ### Author Rebuttal · Reviewer_H4m4 · 2026-04-03
> >
> > Thank you for your detailed responses! I will increase the overall score.

---

### Official Review · Reviewer_v2pC · 2026-03-14

**Soundness:** 2
**Presentation:** 3
**Significance:** 2
**Originality:** 2
**Overall Recommendation:** 4
**Confidence:** 4

**Summary:**

This work points out that one of the limitations of existing strategies or frameworks for evaluating LLM agents' reasoning trajectories is that they often measure the validity of the final outcome without examining the entire trajectory with intermediate steps. Tackling this limitation, the authors propose their evaluation framework, named TRACE, aiming for a more holistic evaluation of tool-using LLM agents. TRACE evaluates each trajectory by producing three metrics, efficiency, hallucination, and adaptivity, by constructing a corresponding "evidence bank." The evidence bank is a set of evidence pieces, each of which is a tuple of (action, action input, observation) for each step. For the efficiency evaluation, it prompts the LLM to identify inefficient (unnecessary) pieces of evidence. For the hallucination evaluation, it lets the LLM determine if the agent's thought at a specific time step is grounded in the evidence available at that point. For the adaptivity evaluation, it uses the LLM to output a score based on whether the agent outputs a reasonable fallback action after observing a failure in its previous attempt to use a tool. In addition, the authors assess the accuracy of the evaluation framework (TRACE) itself, by augmenting the ground-truth trajectories from the GTA and m&m's datasets to form multiple valid trajectories for each task and constructing the corresponding "meta-evaluation" datasets, namely Meta-GTA and Meta-m&m's. Their meta-evaluation results for TRACE (e.g., Table 1 and so on) demonstrate that TRACE provides enhanced accuracy over the naive LLM-as-a-judge approach as well as an existing approach, PIPA, in general.

**Compliance With Llm Reviewing Policy:**

Affirmed.

**Final Justification:**

Their response addresses most of my concerns, and I'm raising my score.

**Key Questions For Authors:**

N/A

**Limitations:**

Yes

**Strengths And Weaknesses:**

**Strengths**

1. The authors perform a fairly comprehensive set of experiments, which backs up the empirical validity of the proposed approach to some degree. This spans from the meta-evaluation of TRACE and its comparison with the baselines (both the naive LLM-as-a-judge approach and PIPA) and the observations with the evaluation of trajectories generated by various LLM agents on the GTA tasks.

2. The problem this work tackles, the evaluation of the full trajectories of LLM agents besides the final answer, has become an important topic in developing reliable LLM agents. Besides just evaluation, such efforts can also be useful for improving LLMs for agentic tasks, by providing training signals, for instance for reinforcement learning fine-tuning.

**Weaknesses**

1. The proposed evaluation metrics, efficiency, hallucination, and adaptivity, are all trajectory-dependent. In other words, the proposed framework, TRACE, does not provide absolute and globally comparable evaluation metrics. While I agree that not relying on ground-truth trajectories is a beneficial trait in some aspects, the lack of reference or baseline materials can make it hard to utilize the evaluation results across different LLM agents or settings and constrain its use cases to local evaluation.

2. The setup of the meta-evaluation of TRACE involves the LLM-based (GPT-4o) augmentation of ground-truth trajectories with the injection of inefficient steps, steps with hallucinations, and adaptive actions, which are the exact three aspects TRACE is designed for. In other words, the meta-evaluation may be too artificial and overfit to the design choices of TRACE. On the other hand, actual trajectories and their suboptimalities generated by various LLM agents can be more diverse and subtle.

3. Given that the proposed evaluation framework, TRACE, mostly relies on prompting LLMs, it still can have non-negligible brittleness or stochasticity in its evaluation results, more so as the tasks of interest get more complex and challenging. While I agree that having a structured evaluation framework helps to mitigate the brittleness of LLM-as-a-judge approaches, the reliance on the underlying LLMs is still a limitation of TRACE. Similarly to naive (i.e., unstructured) LLM-as-a-judge approaches, LLMs that are not strong enough to handle the complexity of tasks may not be very useful as LLM evaluators. The Llama-3.1-8B results in Table 1 suggest this phenomenon, which likely holds for stronger LLMs as the tasks get more challenging.

---

> ### Author Rebuttal · Authors · 2026-03-30
>
> We greatly appreciate the reviewer for the thoughtful assessment and constructive comments on our work. We are particularly grateful that the reviewer recognized our paper poses an important topic in developing reliable LLM agent. In the following, we provide detailed responses to the specific points raised by the reviewer.
>
> ---
>
> ### **W1.** ###
>
> We would like to clarify that trajectory-dependent evaluation is an intentional design choice of TRACE, not a limitation.
>
> For tool-augmented agents, evaluation metrics are and should be inherently trajectory-dependent. Even when two agents produce the same final outcome, the internal quality of their trajectories can differ significantly. While comparing agents against a reference (i.e. golden answer) is the simplest approach for benchmarking agent performance, such metrics fall short of capturing the true quality of an agent’s reasoning process. This is precisely the motivation of our work that trajectory-dependent evaluation is necessary to truly assess agent capabilities, and the design of our metrics reflects this intention.
>
> Our proposed method is **not meant to replace absolute comparison but rather to complement it**. In practice, users can first evaluate agents based on final answer accuracy, and then apply TRACE to conduct a more fine-grained comparison among agents that achieve similar accuracy. This two-stage approach enables researchers and practitioners to measure more detailed performance characteristics and guide the development of agents toward desired properties, such as higher efficiency or lower hallucination rates.
>
> ---
>
> ### **W2.** ###
>
> We acknowledge that the meta-evaluation dataset is constructed in a controlled manner aligned with TRACE’s three evaluation dimensions. However, the purpose of the meta-evaluation is to verify the accuracy of each metric under controlled conditions where the ground-truth labels of those metrics (efficiency, hallucination, adaptivity) are known.
>
> To demonstrate that TRACE also performs well on the more diverse and subtle suboptimalities found in real-world trajectories, we have also conducted human evaluation on the complete set of real trajectories to directly verify the alignment between TRACE’s assessment and human judgements (Agent model: Claude Sonnet, Evaluator model: GPT-4.1). These results, covering every generated sample with high agreement, confirm that TRACE reliably evaluates on naturally generated trajectories.
>
> | |Human agreement|
> |-|-|
> |Efficiency|98.76|
> |Hallucination|95.93|
> |Adaptivity|99.98|
>
> Furthermore, we present case studies in Appendix E, which illustrate that TRACE successfully captures nuanced differences beyond artificially injected patterns.
>
> ---
>
> ### **W3.** ###
>
> We acknowledge that reliance on underlying LLMs is an inherent challenge shared across all LLM-based evaluation approaches, including TRACE. However, we would like to emphasize two important points.
>
> First, a key contribution of TRACE is precisely mitigating the brittleness of LLM evaluators through its structured framework. As shown in Table 1, TRACE consistently improves evaluation accuracy over the naive LLM-as-a-Judge baseline across all model sizes. Notably, for the Llama-3.1-8B model, TRACE achieves approximately a 14% improvement in efficiency evaluation accuracy compared to the unstructured approach. Furthermore, when using an open-source model such as Llama-3.3-70B as the evaluator, TRACE achieves performance comparable to much larger proprietary models (e.g., GPT, Claude, and o3), demonstrating that our structured framework substantially reduces the performance gap between smaller and larger evaluator models. This suggests that **TRACE effectively alleviates the well-known limitation of LLM-as-a-judge approaches**, making reliable evaluation more accessible and cost-effective.
>
> Second, to further validate the reliability of TRACE, we conducted human evaluation using a sample of 200 trajectories of Meta-GTA dataset and present the results below. As expected, human evaluators achieve higher accuracy than TRACE, but the **human evaluation required significantly more time, and this cost increased substantially as trajectories became longer**. Given that the performance gap between TRACE and human evaluators is relatively modest, we believe this **trade-off between a slight reduction in accuracy and a substantial gain in scalability and cost-efficiency is well justified**, particularly for large-scale evaluation scenarios where human annotation is prohibitively expensive. This finding underscores the essential need for an effective LLM-based evaluator and serves as further evidence validating our core contribution.
>
> |               | LLM-as-a-judge | TRACE | Human evaluation |
> |-----|---|-|-|
> | Efficiency    | 86.08          | 94.64 | 98.38            |
> | Hallucination | 89.68          | 95.21 | 97.96            |
> | Adaptivity    | 98.83          | 99.63 | 100              |

---

> > ### Author Rebuttal · Reviewer_v2pC · 2026-04-04
> >
> > I thank the authors for the comprehensive, detailed response.
> >
> > Their response addresses most of my concerns, and I'm raising my score.

---

### Decision · Program_Chairs · 2026-04-30

**Decision:**

Accept (regular)

**Comment:**

This paper proposes TRACE, a reference-free evaluation framework for tool-augmented LLM agents that assesses reasoning trajectories across three dimensions (efficiency, hallucination, and adaptivity) via an incrementally constructed evidence bank. The core idea of this work is that, structuring the agent's interaction history as a typed evidence bank provides the LLM evaluator with focused, step-local context sufficient to detect inefficiencies, hallucinations, and failures of adaptivity without relying on any pre-defined ground-truth trajectory. The authors further introduce two meta-evaluation datasets, Meta-GTA and Meta-m&m's, constructed by augmenting existing benchmarks with synthetically injected flaws and corresponding labels, and show that TRACE outperforms both a vanilla LLM-as-a-judge baseline and PIPA across a range of evaluator models, including small open-source ones.

The post-rebuttal consensus is clearly positive. Three assigned weak accept and one assigned accept. After rebuttal, reviewers v2pC, H4m4, and sLwe each reported their concerns as fully resolved and raised their scores accordingly; reviewer hQMh maintained a positive recommendation, while requesting three specific revisions for the camera-ready version: (1) a discussion of the efficiency–adaptivity trade-off; (2) an explicit justification of the hallucination definition; and (3) incorporation of the supplementary experiments described in the rebuttal. These are tractable, well-scoped requests that require no structural changes to the paper's central claims. I expect the authors to fulfill these commitments.

Overall, TRACE is a solid, well-motivated contribution to the emerging area of process-level agent evaluation. The work is technically sound, the experiments are reasonably comprehensive, and the observations from the applied GTA analysis are of practical interest to the community. The weaknesses noted above are real but do not undermine the paper's core findings. Conditional on the authors implementing the revisions requested by reviewer hQMh, I recommend Accept.